# Genome mining unveils a class of ribosomal peptides with two amino termini

Hengqian Ren [1,2,8], Shravan R. Dommaraju [2,3,8], Chunshuai Huang [2], Haiyang Cui[2], Yuwei Pan[4], Marko Nesic [2,3], Lingyang Zhu [5], David Sarlah [2,3], Douglas A. Mitchell [2,3,6] ✉ & Huimin Zhao [1,2,3,7] ✉

The era of inexpensive genome sequencing and improved bioinformatics tools has reenergized the study of natural products, including the ribosomally synthesized and post-translationally modified peptides (RiPPs). In recent years, RiPP discovery has challenged preconceptions about the scope of post-translational modification chemistry, but genome mining of new RiPP classes remains an unsolved challenge. Here, we report a RiPP class defined by an unusual (S)-$N_2,N_2$-dimethyl-1,2-propanediamine (Dmp)-modified C-terminus, which we term the daptides. Nearly 500 daptide biosynthetic gene clusters (BGCs) were identified by analyzing the RiPP Recognition Element (RRE), a common substrate-binding domain found in half of prokaryotic RiPP classes. A representative daptide BGC from *Microbacterium paraoxydans* DSM 15019 was selected for experimental characterization. Derived from a C-terminal threonine residue, the class-defining Dmp is installed over three steps by an oxidative decarboxylase, aminotransferase, and methyltransferase. Daptides uniquely harbor two positively charged termini, and thus we suspect this modification could aid in membrane targeting, as corroborated by hemolysis assays. Our studies further show that the oxidative decarboxylation step requires a functionally unannotated accessory protein. Fused to the C-terminus of the accessory protein is an RRE domain, which delivers the unmodified substrate peptide to the oxidative decarboxylase. This discovery of a class-defining post-translational modification in RiPPs may serve as a prototype for unveiling additional RiPP classes through genome mining.

Ribosomally synthesized and post-translationally modified peptides (RiPPs) constitute a major family of natural products present in all domains of life[1,2]. RiPP biosynthesis starts with the expression of a ribosomal precursor peptide, usually consisting of an *N*-terminal leader region and a *C*-terminal core region (Fig. 1). The leader region

recruits the biosynthetic enzymes that perform post-translational modifications (PTMs) on the core region. Ultimately, the leader region is released from the modified core, often by a peptidase, to yield the final RiPP product. Due to the high variability in the sequences of precursor peptides and the growing list of chemistries performed by

[1]Department of Chemical and Biomolecular Engineering, University of Illinois at Urbana-Champaign, Urbana, IL, USA. [2]Carl R. Woese Institute for Genomic Biology, University of Illinois at Urbana-Champaign, Urbana, IL, USA. [3]Department of Chemistry, University of Illinois at Urbana-Champaign, Urbana, IL, USA. [4]Department of Molecular and Cellular Biology, University of Illinois at Urbana-Champaign, Urbana, IL, USA. [5]School of Chemical Sciences, NMR Laboratory, University of Illinois at Urbana-Champaign, Urbana, IL, USA. [6]Department of Microbiology, University of Illinois at Urbana-Champaign, Urbana, IL, USA. [7]Department of Bioengineering, University of Illinois at Urbana-Champaign, Urbana, IL, USA. [8]These authors contributed equally: Hengqian Ren, Shravan R. Dommaraju. ✉e-mail: douglasm@illinois.edu; zhao5@illinois.edu

the modification enzymes, RiPPs exhibit vast structural diversity and hence a broad range of biological functions. Unlike non-ribosomal peptides, which have large genomic footprints to encode the required megasynthetases, RiPPs are genomically compact, and their ribosome-dependent biosynthetic logic endows RiPPs with attractive engineering potential[2–5]. In addition, the frequently observed promiscuous activity of biosynthetic enzymes also brings particular interest to the characterization of biosynthetic gene clusters (BGCs) that produce RiPPs with unknown PTMs[6,7].

Despite the success of genome mining, its direct application in RiPP discovery remains challenging, principally owing to the lack of a ubiquitous feature across RiPP biosynthetic pathways and low accuracy in precursor peptide prediction[8]. Among the 48 classes of RiPPs reported to date, nearly all of the founding members of each class were identified by bioassay-guided screening, and their ribosomal origin was only later identified[2]. While bioassay-guided isolation is a powerful discovery strategy, known shortcomings include high rediscovery rates, inability to identify compounds that differ in activity from those examined by the bioassay, and challenges in specifically targeting natural products from cryptic BGCs[9,10]. Therefore, significant effort has been spent to develop bioinformatics tools for RiPP genome mining, using both class-dependent and class-independent strategies[8].

Broadly speaking, class-dependent RiPP genome mining relies on sequence homology to well-characterized biosynthetic enzymes, affording BGC predictions with high fidelity but potentially reduced novelty of chemical structures. New RiPP classes, such as the pearlins[11,12], ranthipeptides[13], spliceotides[14], ryptides[15], rotapeptides[16], and tryglysins[17,18] were discovered using this strategy, but the scope of these genome mining efforts was limited to lanthipeptide dehydratases and radical S-adenosylmethionine (SAM)-dependent enzymes that were already well established in RiPP biosynthesis. In addition, key RiPP biosynthetic enzymes, such as cytochrome P450 enzymes for cittilins[19], atropitides[20], and biarylitides[21], or radical SAM enzymes for additional classes, are widely distributed outside of RiPP BGCs; therefore, the presence of genes encoding such enzymes cannot be used as a reliable RiPP biomarker. In contrast, class-independent genome mining tools have also been developed in recent years using features common across natural product classes, but these have yet to deliver new enzyme families to RiPP biosynthesis[10].

One unexploited approach to identify first-in-class RiPPs is to leverage the RiPP precursor recognition element (RRE), a domain responsible for recruiting the precursor peptide to RiPP biosynthetic enzymes[22–24]. As the most common domain involved in RiPP biosynthesis, the RRE has been found in 19 of the 41 known prokaryotic RiPP classes, and its frequency suggests it could be used as a class-independent handle for RiPP genome mining. The prevalence of the RRE is obscured by its small size and high sequence variability, but the recent report of RRE-Finder, a bioinformatics tool that rapidly and accurately detects RRE domains, offers the potential to discover first-in-class RiPP BGCs by identifying high-confidence RRE domains from public genome databases[23] (Fig. 1).

In this work, we employ RRE-Finder[23] and RODEO[25] in a class-independent manner to uncover a RiPP class. The identified BGCs are unlike any other reported RiPP and ubiquitously encode a domain of unknown function (DUF)-RRE fusion protein, oxidative decarboxylase, aminotransferase, and methyltransferase. Direct cloning and heterologous expression of a representative BGC from *Microbacterium paraoxydans* DSM 15019 yields peptides containing a native N-terminus and an (S)-$N_2$,$N_2$-dimethyl-1,2-propanediamine (Dmp)-modified C-terminus. These peptides are further identified as synergistic hemolysins, with their bioactivity putatively conferred by helical structure and net positive charge on both termini. Study of the biosynthetic pathway indicates that Dmp is derived from the invariant C-terminal Thr of the precursor peptide through successive oxidative decarboxylation, transamination, and dimethylation. Subsequent

characterization shows that the oxidative decarboxylase is leader peptide-dependent and tolerates variability in the core region. Overall, the discovery of daptides [(S)-$N_2$,$N_2$-dimethyl-1,2-propanediamine-containing peptides] showcases class-independent genome mining for RiPP biosynthetic enzyme families and provides a logical route for nature to produce ribosomal peptides with two amino termini.

## Results

### RRE domains guide RiPP class discovery

RiPP genome mining has primarily focused on known RiPP-modifying enzymes for new BGCs[2]. While this approach continues to yield new biosynthetic pathways, the discovery potential is restricted to known enzyme families. To uncover RiPP classes independent of an established chemistry, we devised an orthogonal approach using the RRE domain[22]. Initial analysis of all identified RRE domains faced two main challenges: (i) the large number of RRE-containing proteins, and (ii) the lack of sequence similarity between disparate RRE-containing proteins. These features of RRE analysis made traditional genome mining approaches computationally difficult and not generalizable across all RRE-containing BGCs. To circumvent these challenges, a workflow was devised to leverage the co-occurrence of RRE domains with nearby encoded open-reading frames (ORFs) using the genome mining tool RODEO (Supplementary Fig. 1, Supplementary Note)[25]. All RRE-containing proteins predicted in the RRE-Finder exploratory mode dataset were analyzed by RODEO for local hypothetical short ORFs and gene co-occurrence[23]. RRE-containing proteins were then subjected to all-by-all BLASTP and sorted by sequence similarity into large RRE families. Each RRE family was recursively analyzed for highly similar co-occurring ORFs at a series of bitscore similarity thresholds. Using this approach, RRE families with highly similar co-occurring ORFs were generated without requiring any visualization of phylogenetic data or a uniform similarity threshold (Supplementary Fig. 1).

Output RRE families were analyzed, and one set of BGCs was selected for further characterization based on the uniqueness of the co-occurring genes. The selected RRE domains were N-terminally fused to an intramembrane site-2 protease. A sequence logo of the co-occurring ORFs was generated, showing a conserved N-terminal motif (ELExMEAP), a stretch of residues rich in branched-chain aliphatic amino acids, and an invariant C-terminal Thr (Fig. 1, Supplementary Fig. 2). This approach identified 184 short ORFs encoded near 80 RRE-peptidase fusion proteins (average of 2.3 ORFs/BGC). The encoding of multiple similar short ORFs within each BGC increased confidence in their prediction as precursor peptides, and the position of the N-terminal motif within the putative leader region suggested it may function as the recognition sequence for the associated biosynthetic enzymes. Initial gene co-occurrence analysis identified multiple highly co-occurring genes encoding multi-component ABC transporters, a pyridoxal phosphate (PLP)-dependent aminotransferase, an NAD(P)-dependent alcohol dehydrogenase, and a SAM-dependent methyltransferase (Fig. 1, Supplementary Table 1). Each BGC also contained an additional gene, which lacks sequence similarity to any known protein. However, structural similarity analysis using HHpred[26] predicted a C-terminal RRE domain, suggesting its inclusion in the potential biosynthetic pathway (Supplementary Fig. 3).

We next sought to expand the putative class by gathering all predicted BGCs from the NCBI non-redundant database. To gather a comprehensive set of putative modifying enzymes, Position Specific Iterative-Basic Local Alignment Search Tool (PSI-BLAST) methods were applied to each predicted biosynthetic protein, and the resulting BGCs were compiled. The originally identified BGCs of this class were restricted to Actinomycetota, but PSI-BLAST searches using the DUF-RRE fusion protein as a query returned a small number of similar BGCs from the phylum Bacillota. A profile Hidden Markov Model (pHMM) was constructed from the 184 initially predicted precursor peptides from Actinomycetota and used to identify a BGC from *Bacillus cereus*

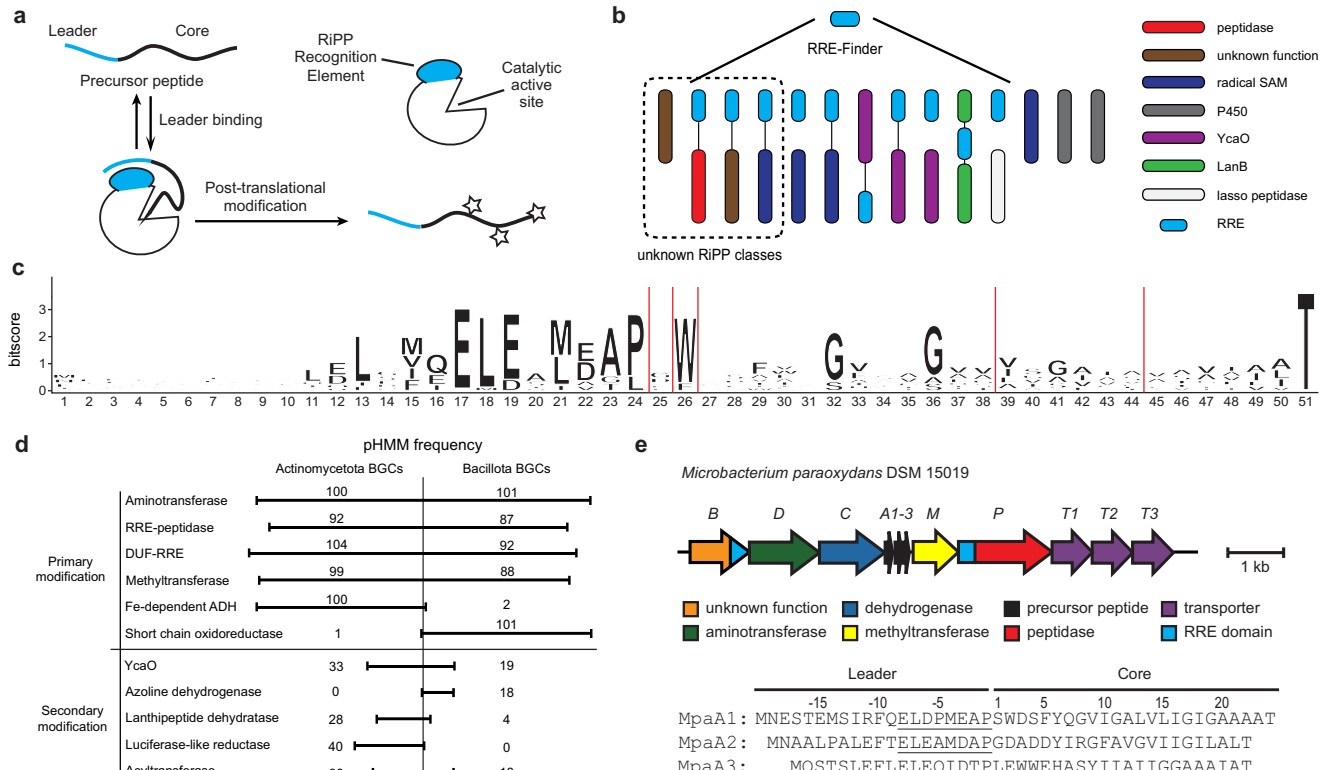

**Fig. 1 | RRE-based discovery of a RiPP class. a** General RRE-dependent RiPP biosynthetic pathway. **b** Stylized depiction for use of RRE-Finder to discover RiPP BGCs independent of known RiPP chemistry. **c** Daptide precursor peptide sequence logo (*n* = 184). **d** Frequency of common pHMM hits in daptide BGCs. Common best hit pHMMs were counted and grouped as follows: Aminotransferase – PF00202, TIGR00508, TIGR00707; RRE-peptidase – PF02163, Actino_DapP_RRE, Bacill_DapP_RRE; DUF-RRE – Actino_DapB_RRE, Bacill_DapB_RRE; Methyltransferase –

PF13649, PF08241, PF13849, PF08242; Fe-dependent alcohol dehydrogenase (ADH) – PF00465, TIGR03405, PF13685; Short chain oxidoreductase – PF13561, TIGR01830, TIGR02638; YcaO – PF02624, TIGR01575; Azoline dehydrogenase – TIGR03605; Lanthipeptide dehydratase – PF00069, TIGR03897; Luciferase-like reductase – TIGR03564, PF00296; Acyltransferase – PF00583, TIGR01575.
**e** Targeted BGC from *Microbacterium paraoxydans* DSM 15019 identified from the genome mining analysis of RREs.

strain B4082 (Supplementary Fig. 4, Supplementary Data 1). PSI-BLAST searches were conducted using the *B. cereus* DUF-RRE fusion protein as a new query to expand the number of Bacillota BGCs. A phylogenetic comparison of the BGCs revealed both similarities and differences. One difference pertains to the number of precursor peptides encoded per BGC, with some Bacillota BGCs encoding >10 precursor peptides (Supplementary Fig. 5). Actinomycetota BGCs also encode a predicted Fe-dependent alcohol dehydrogenase (Pfam[27] identifier PF00465), while Bacillota BGCs encode a different class of oxidoreductase (PF13561) and a shorter protease domain (Supplementary Fig. 6). Despite similar RiPP contexts, biosynthetic proteins from Actinomycetota were typically as similar to characterized outgroup proteins as they were to their Bacillota counterparts (Supplementary Fig. 7, phyloXML data is provided as Supplementary Data 2). In total, 1441 putative precursor peptides were identified from 483 BGCs (Supplementary Data 1, Actinomycetota – 791 precursor peptides in 302 BGCs; Bacillota – 647 precursor peptides in 180 BGCs; Pseudomonadota – 3 precursor peptides in 1 BGC). In general, there is little conservation of gene order or precursor peptide number within these BGCs. Several potential secondary modifying genes were identified, such as YcaO proteins, LanK$_C$/LanJ$_C$ pairs, glycosyltransferases, and nucleotidyltransferases (Fig. 1, Supplementary Data 1, Supplementary Table 1). This class of BGCs combines a minimal set of primary modifying enzymes, as well as secondary (ancillary) tailoring enzymes from other classes, suggesting the final biosynthetic transformations may work in concert in other RiPP pathways (Supplementary Fig. 4). After filtering the dataset for available strains and for BGCs encoding only primary modifying enzymes, we chose the representative from

*Microbacterium paraoxydans* DSM 15019 for experimental characterization (Fig. 1).

## Discovery of peptides with Dmp-modified *C*-termini

To examine the predicted BGC, we directly cloned the representative BGC from *Microbacterium paraoxydans* DSM 15019 (*mpa*) using the recently developed CAPTURE (Cas12a-assisted precise targeted cloning using in vivo Cre-lox recombination) method[28]. Besides genes in the predicted BGC, flanking genes were also included in the cloning region to ensure successful expression (Supplementary Fig. 8). The cloned *mpa* and the pBE45 empty vector were then individually conjugated[29] into *Streptomyces lividans* TK24 and *Streptomyces albus* J1074. After cultivation for 5 d, colonies were picked, extracted with methanol, and analyzed by matrix-assisted laser desorption/ionization-time-of-flight mass spectrometry (MALDI-TOF-MS) (Fig. 2, Supplementary Fig. 9). A series of ions only observed in strains transformed with *mpa* corresponded to a loss of 17 Da from the three predicted core regions (MpaA1-A3), with proteolysis *C*-terminal to the conserved Pro (Fig. 1). This consistency suggested that the three products shared the same post-translational modification(s). Larger scale cultures in liquid media were then prepared to afford the material required for structural characterization. After growth for 5 d at 30 °C, bacteria were collected by centrifugation, metabolites extracted with MeOH, and purified by solid-phase extraction and HPLC (Supplementary Fig. 10).

The HPLC-purified products were analyzed by high-resolution mass spectrometry and tandem mass spectrometry (HR-MS/MS). Analysis of the collision-induced dissociation spectra indicated that

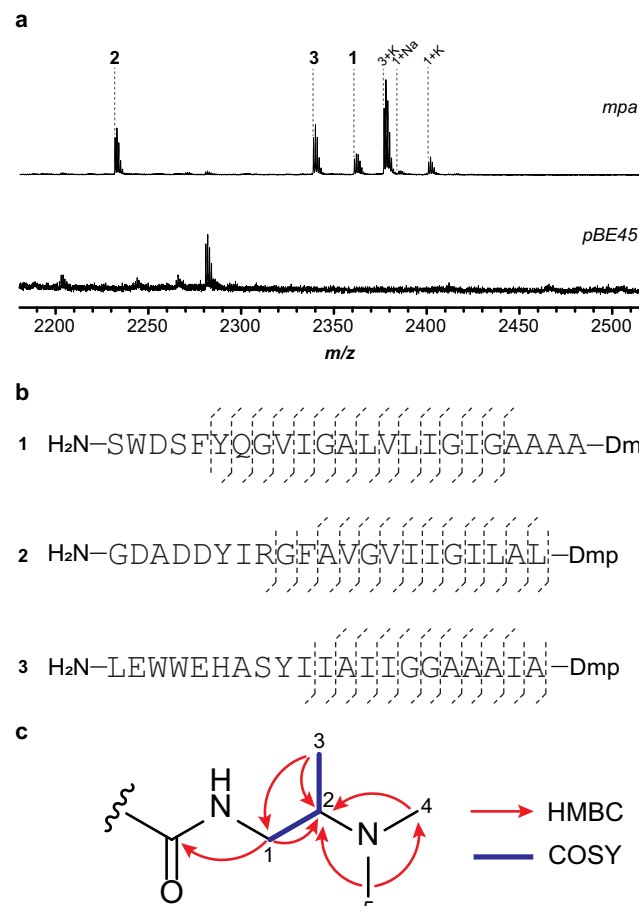

**Fig. 2 | Heterologous expression and product characterization of *mpa*.**
**a** MALDI-TOF mass spectra of the methanol extracts of *S. albus* J1074 containing *mpa* (**1**, *m/z* 2363; **2**, *m/z* 2232; **3**, *m/z* 2339) and the pBE45 empty vector. Sodium and potassium ions are denoted as '+Na' and '+K'. **b** Summary of the HR-MS/MS results for compound **1-3**. **c**, key HMBC and COSY correlations for the Dmp moiety in **1**. NMR spectra of **1** and assigned signals of Dmp are shown in Supplementary Fig. 14 and Supplementary Table 2, respectively.

the 17 Da mass loss of modified MpaA1-A3 was confined to the *C*-terminal Thr (Fig. 2, Supplementary Fig. 11-13). Considering the predicted functions of genes encoded by the *mpa* BGC, we proposed that Thr underwent conversion to Dmp. To evaluate this hypothesis, a sufficient amount of **1** was accumulated for structure determination, due to its higher post-purification yield than **2** and **3**. The peptide was only sparingly soluble in commonly used NMR solvents (H₂O, methanol, acetone, acetonitrile, chloroform, DMSO, and pyridine) but was highly soluble in hexafluoroisopropanol (HFIP) which is often used to dissolve aggregated proteins and peptides[30]. NMR analyses (¹H, ¹³C, ¹H-¹H COSY, ¹H-¹³C HSQC and ¹H-¹³C HMBC) were then conducted in HFIP-$d_2$ and a substructure corresponding to Dmp was deduced. The HMBC correlations from the *N*-methyl (H₃-4 and H₃-5) to C-2 showed the attachment of a dimethylamino group to C-2. Other data, including the HMBC correlations from the H₃-3 to C-1 and C-2, the HMBC correlation from H₂-1 to C-2, and the COSY data, revealed the C-1 to C-3 Dmp substructure. The amide bond between Dmp and Ala-20 is suggested by the HMBC correlation from H₂-1 in Dmp, as well as H₁-1' and H₃-2' in Ala-20, to the carbonyl carbon ($\delta_C$ = 175.8) (Fig. 2, Supplementary Fig. 14, Supplementary Table 2, Supplementary Table 3).

Extensive signal overlap was observed in the NMR spectra of **1**, so orthogonal methods were employed to confirm the structure of **1**. Accordingly, **1** was hydrolyzed and derivatized by Marfey's reagent[31] to determine the identity of Dmp and the remaining residues present.

While the amino acid standards were obtained commercially, authentic samples of (*R*)-Dmp and (*S*)-Dmp were chemically synthesized (Supplementary Fig. 15). Using an LC-MS, all assignable amino acids (Ser, Asp, Phe, Gln, Val, Ile, Leu, Ala, and Tyr) were determined to be in the proteinogenic (*S*)-configuration (Supplementary Fig. 16). UHPLC afforded sufficient separation of the derivatized (*R*)-Dmp and (*S*)-Dmp standards, with subsequent co-elution with hydrolyzed **1** confirming the presence of (*S*)-Dmp (Supplementary Fig. 17).

To further validate Dmp as the primary and class-defining PTM, a second BGC identified from *Streptomyces capuensis* NRRL B-3501 (*sca*) was selected for experimental characterization. In addition to the primary modifying enzymes, the *sca* BGC also encodes a LanK$_C$/LanJ$_C$ pair that was predicted to convert *L*-Ser to *D*-Ala via a dehydroalanine intermediate[32-34] (Supplementary Fig. 18). The *sca* BGC underwent direct cloning by the CAPTURE method[28], heterologous expression in *S. albus* J1074, and MALDI-TOF-MS analysis (Supplementary Fig. 19). The most prominent ion observed from the *sca*-transformed strain was fragmented using MALDI LIFT-TOF/TOF MS and confirmed to derive from ScaA2. The corresponding fragmentation pattern confirmed the *C*-terminus had been converted to Dmp and the five *L*-Ser residues had been converted to Ala (Supplementary Fig. 19). Marfey's assay performed on a mixture of the two major products further showed the peptides contained *D*-Ala residues and (*S*)-Dmp (Supplementary Fig. 20). With multiple characterized BGCs producing Dmp-modified peptides, we anticipate that the Dmp modification will be conserved across the RiPP class, hereafter termed daptides.

## Examination of daptide bioactivity

As **1-3** contain regions of hydrophobic residues and showed limited solubility in most solvents, we theorized that **1-3** might form hydrophobic α-helical secondary structures. Secondary structure prediction of MpaA1-A3 and circular dichroism spectra of **1-3** support adoption of an α-helical conformation (Supplementary Fig. 21). Melittin, the major pain-producing component of honeybee venom, disrupts membranes via its hydrophobic α-helical structure and a group of positively charged residues near the amidated *C*-terminus (Supplementary Fig. 22), giving it additional antimicrobial and hemolytic activity. Given the positive charge at the (former) *C*-terminus of **1-3**, we hypothesized that daptides may imitate this strategy to interact with membranes[35]. Daptides **1-3** were first analyzed for antimicrobial activity by standard agar diffusion assay, but no significant growth inhibition was observed for any of the tested Bacillota, Actinomycetota, Pseudomonadota, and Ascomycota strains (Supplementary Table 4). To further evaluate their potential membrane interactions, **1-3** were incubated with bovine erythrocytes and exhibited hemolytic activity (Supplementary Fig. 23). Because daptide BGCs mostly encoded multiple precursors (Supplementary Fig. 5), combinations of **1-3** were also examined for hemolytic activity. Some combinations (i.e., **1 + 3**, **2 + 3**, and **1 + 2 + 3**) exhibited higher hemolytic activity than the individual peptides alone, underscoring a collaborative bioactivity (Supplementary Fig. 23).

## Biosynthesis of Dmp via a three-step pathway

To determine the minimal set of genes responsible for daptide **1-3** biosynthesis, a series of gene omissions in the *mpa* BGC (Supplementary Fig. 24) were constructed by the DNA assembler method[36] and conjugated into *S. albus* J1074 for expression. The colony extracts were analyzed by MALDI-TOF-MS, which indicated that *mpaABCDMP* were sufficient for synthesizing the fully modified peptides, albeit with compromised signal intensity compared to the *mpa* BGC, suggesting other flanking genes may enhance productivity. No diagnostic intermediates were observed when conserved biosynthetic genes were omitted (Supplementary Fig. 24). Therefore, we elected to express the *mpaABCDM* pathway in *E. coli*. Genes were codon-optimized, commercially synthesized, and refactored using a previous method (Supplementary Table 5, Supplementary Table 6)[37]. The precursor peptides

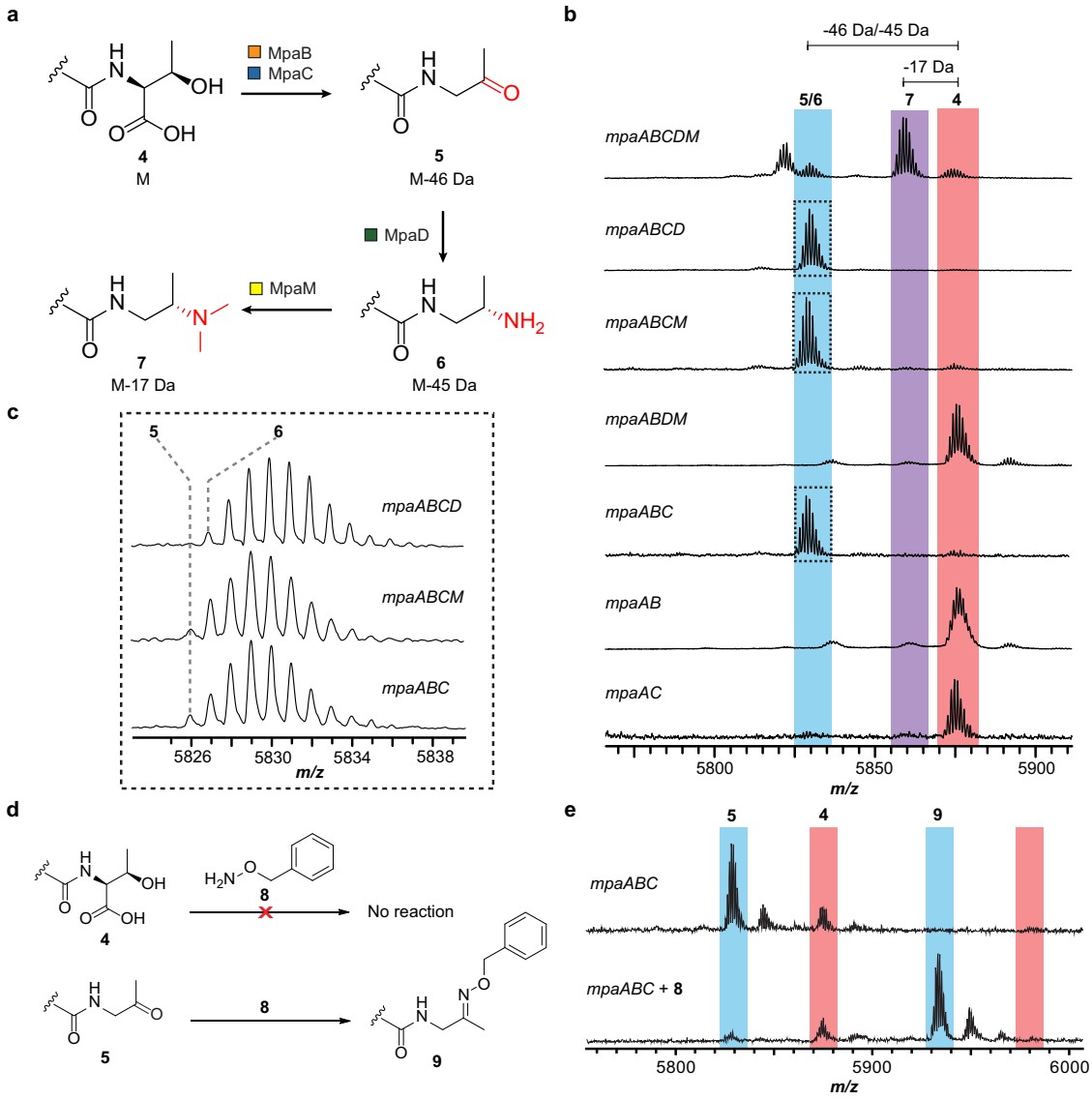

**Fig. 3 | Characterization of the biosynthetic pathway. a** Proposed biosynthetic route for the Dmp modified *C*-terminus. **b** MALDI-TOF mass spectra of IMAC-purified MpaA1 that were co-expressed with the indicated Mpa biosynthetic proteins in *E. coli* (**4**, *m/z* 5872; **5**, *m/z* 5826; **6**, *m/z* 5827; **7**, *m/z* 5855). **c** Enlarged MALDI-TOF mass spectra of *mpaABCD*, *mpaABCM*, and *mpaABC* co-expressions in panel **b**. **d** Carbonyl reactivity using aminooxy probe **8**. **e** MALDI-TOF MS results for the unmodified peptide **4** and ketone intermediate **5** (oxime **9**, *m/z* 5931).

(MpaA1-A3) were individually *N*-terminal His-tagged to facilitate purification by immobilized metal affinity chromatography (IMAC). The purified products were then further desalted using C18 ZipTip and analyzed by MALDI-TOF-MS. During optimization of the expression conditions, we observed that the signals of fully modified products were significantly improved when using M9 medium with extended cultivation and co-expression of GroES/EL chaperones (Supplementary Fig. 25-28)[38]. MpaA1 routinely delivered the most intense MS peaks; therefore, the following experiments were carried out using *mpaA1* exclusively (hereafter, *mpaA*).

A biosynthetic scheme was next proposed, employing each critical gene in a distinct step in the conversion of a *C*-terminal Thr to Dmp (Fig. 3). The unmodified peptide **4**, containing an intact *C*-terminal Thr would undergo oxidative decarboxylation to ketone **5**, followed by transamination to yield primary amine **6**, and finally dimethylation to afford the final tertiary amine-containing Dmp **7**. One advantage of a refactoring strategy is the opportunity to evaluate biosynthetic intermediates by rapidly generating gene omission

constructs. Thus, we evaluated the roles of *mpaBCDM* in the proposed biosynthetic route with a panel of gene omissions. Expression of *mpaABCD* (i.e., omission of *mpaM*, predicted involvement in the last step) resulted in the accumulation of a product 45 Da lighter than starting peptide **4**. This result was consistent with primary amine **6** and concurred with the expected role of MpaM in dimethylation. Expression of *mpaABCM* or *mpaABC* resulted in the accumulation of a product 46 Da lighter than starting peptide **4**, consistent with ketone **5** and a transaminase activity for MpaD. Given the mass similarity to amine **6**, and to further support the existence of ketone **5** as a biosynthetic intermediate, we employed the use of aminooxy probe **8**, which would yield an oxime after reaction with aldehyde and ketone functional groups[39]. The MS data revealed that **5** was converted to oxime **9**, whereas **4** was unreactive under identical conditions (Fig. 3). Expression of *mpaABDM*, *mpaAB*, and *mpaAC* constructs yielded only unmodified starting peptide **4**, which taken together implicated MpaB (DUF-RRE fusion) and MpaC (annotated alcohol dehydrogenase) in the first reaction, oxidative decarboxylation of the *C*-terminal Thr.

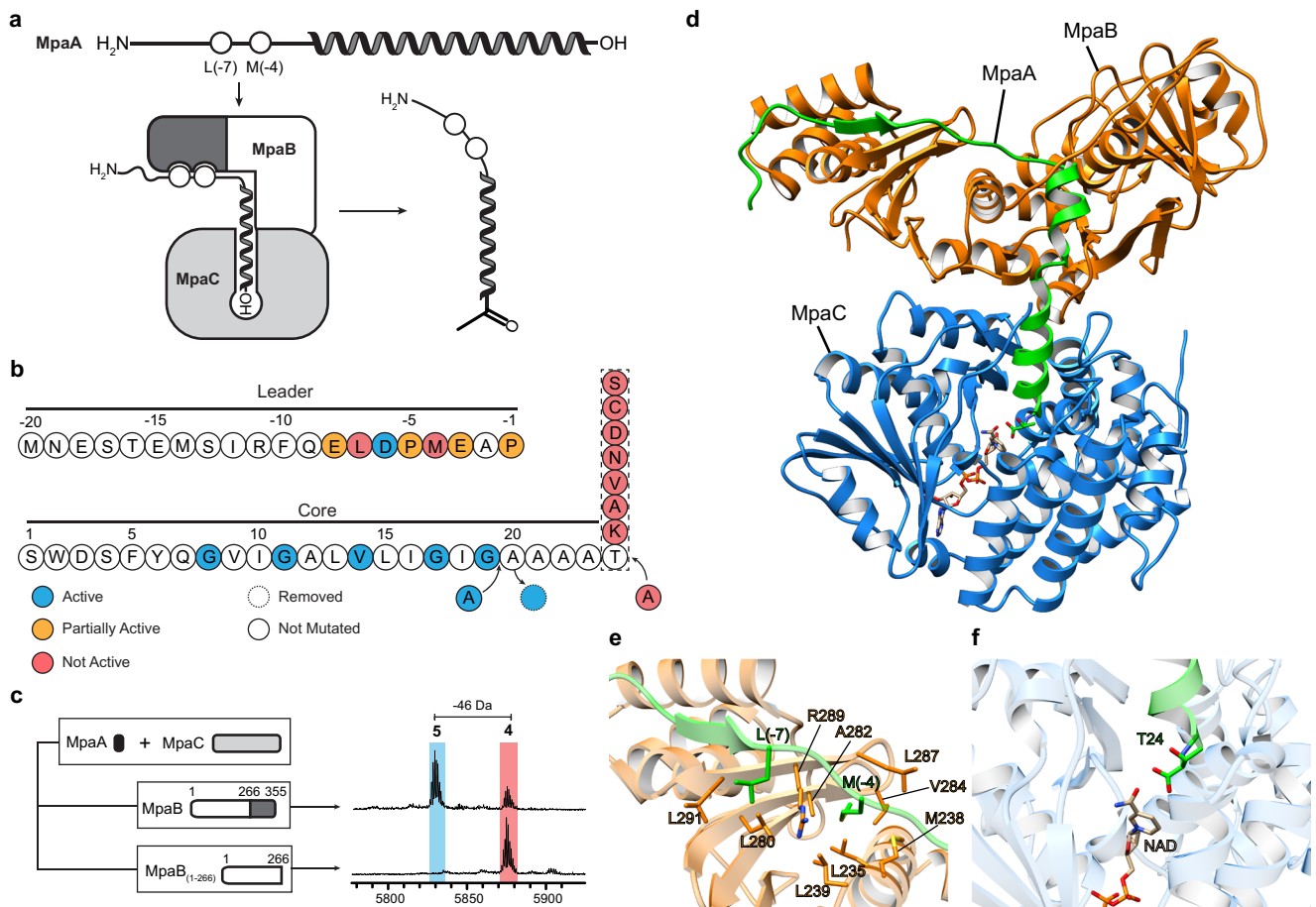

**Fig. 4 | Characterization of ketone intermediate formation by MpaB and MpaC.**
**a** Proposed interaction model for MpaA, MpaB, and MpaC. The RRE domain (dark grey) of MpaB interacts with the conserved leader region of MpaA. The C-terminus (depicted as "-OH") of MpaA is directed into the active site of MpaC for modification. **b** Summary of MpaA variant processing upon *E. coli* co-expression with MpaB/C. **c** Mass spectra of MpaA after co-expression with MpaC, and MpaB or MpaB$_{1-266}$. The RRE domain of MpaB is dark grey. **d** The structure of MpaA1-MpaB-MpaC predicted using AlphaFold-Multimer[40]. MpaA1, MpaB, and MpaC are shown as cartoons in green, orange, and blue, respectively. The positioning of cofactor NADP was imported using a homologous alcohol dehydrogenase (PDB ID: 6C76) as a template. **e**, View of the leader peptide-RRE domain binding interface[40]. Select residues of MpaA and MpaB are in green and orange, respectively. **f** View of the C-terminal Thr24 of MpaA in the substrate-binding pocket of MpaC.

## Oxidative Decarboxylation of the C-terminal Thr

The gene omission studies showed that ketone **5** can only form in the presence of MpaB and MpaC. Although the RRE domain fused to the peptidase was used to identify the daptide BGCs, MpaB also contains a predicted RRE domain (Supplementary Fig. 3). Therefore, we hypothesized that the MpaB RRE domain might bind the MpaA1-A3 leader region and deliver the C-terminal Thr to MpaC for modification. To evaluate this interaction, Ala substitutions were introduced into the conserved motif of the MpaA1 (i.e., MpaA) leader region, and the resulting products after co-expression with MpaB and MpaC were analyzed by MALDI-TOF-MS (Fig. 4, Supplementary Fig. 29). The D(-6)A variant was processed at a level equal to wild-type, while E(-8)A, P(-5)A, E(-3)A, and P(-1)A yielded a reduced level of product formation. Two variants, L(-7)A and M(-4)A, were devoid of detectable oxidative decarboxylation. Removal of the C-terminal RRE domain from MpaB (i.e., expression of MpaB$_{1-266}$), also was incapable of generating ketone **5**, further validating a specific interaction between the leader region of MpaA and the RRE domain.

To assess the substrate tolerance of MpaC, the C-terminal Thr of MpaA was replaced with Ser, Cys, Asp, Asn, Val, Ala, and Lys. No mass change was observed for any variant, indicating a stringent selectivity for Thr (Fig. 4, Supplementary Fig. 30). Another MpaA variant, T24insA, appended Ala to the C-terminus and was not a substrate for MpaC (Supplementary Fig. 31). However, variants that increased or decreased the length of the core region (i.e., A20dup and A20del) were readily tolerated along with all other tested substitutions (Supplementary Fig. 32). Thus, while MpaC is highly chemoselective for a C-terminal Thr, the exact positioning of the C-terminal Thr and the identity of the intervening residues are not critical. A high tolerance to core peptide variation was anticipated from the bioinformatics analysis, as high variation of the core peptide sequences was observed even for those within the same BGC, while the C-terminal residue remained a Thr across the daptide class (Fig. 1).

To shed more light on the formation of ketone **5**, AlphaFold-Multimer, a recently reported structure prediction tool for protein complexes, was used to calculate the protein-protein interactions between MpaA, MpaB, and MpaC[40]. MpaC is predicted to use NAD(P) as a cofactor, so a structural alignment was used to model NADP from a homologous alcohol dehydrogenase, given the current inability of AlphaFold to predict ligands. The complex with NADP was then relaxed to yield a final predicted structure (Fig. 4, Supplementary Fig. 33). These results aligned well with our hypothesis that MpaB recognizes the leader region of MpaA and recruits MpaC to form ketone **5**. Two leader region residues of MpaA, Leu(-7) and Met(-4), putatively interact with MpaB in adjacent hydrophobic pockets, supporting the experimental findings that ketone **5** formation was abolished with the L(-7)A and M(-4)A variants (Fig. 4). Structural and electrostatic potential analysis of the predicted MpaB-MpaC interface shows a high degree of

charge and shape complementarity (Supplementary Fig. 34). Finally, the Alphafold prediction directs Thr24 toward the active site NADP at an appropriate distance for catalysis.

## Discussion

With recent advances in DNA sequencing and bioinformatics, natural products with diverse structures and bioactivities have been continuously uncovered[41]. RiPPs constitute one of the major classes of natural products, and the number of characterized compounds has grown rapidly in the last decade. However, unveiling new PTMs remains a major challenge in the genome mining of RiPPs, as class-independent genome mining methods have only recently received broad attention from the field. With RRE domains being present in 50% of prokaryotic RiPP classes, we saw an opportunity to exploit the RRE as a RiPP biomarker. In this work, RRE domains identified by RRE-Finder were binned into families, guided by co-occurrence with nearby short ORFs. Employing this method led to the discovery of the daptide class from a group of RRE-containing peptidases. Enabled by synthetic biology methods for direct cloning and refactoring, pathway characterization revealed the biosynthetic route to Dmp. Given the generality of the methods used in this study, the success of the daptide discovery could be replicated for yet-undiscovered RiPPs.

To assess the frequency of Dmp-modified C-termini in known compounds, we performed a substructure search using Dmp with several adjoining peptide bonds on SciFinder (https://scifinder-n.cas.org/). This search identified hominicin, a peptidic natural product produced by *Staphylococcus hominis* MBBL 2-9[42]. Besides a Dmp-modified C-terminus, hominicin also contains several dehydrobutyrines and a dimethylated N-terminus. Unfortunately, the genome of *S. hominis* MBBL 2-9 is not available, and no sequenced *S. hominis* strains contain a daptide-producing BGC. However, the predicted precursor peptide of hominicin is virtually identical to a precursor peptide identified from *Staphylococcus pseudintermedius* strain B32 (Supplementary Fig. 35). The *S. pseudintermedius* BGC contains DapBCDM homologs (Supplementary Table 7), a second methyltransferase, and a lanthipeptide dehydratase, likely accounting for the additional methylations and dehydrobutyrines observed in hominicin. Retrospectively, these similarities identify hominicin as a daptide. Hominicin is growth suppressive towards *Staphylococcus aureus*, an activity which was not observed for daptides 1-3.

In comparison to the hominicin example, genome mining was critical for the discovery of daptides 1-3 and offers leads for the discovery of further compounds. In our bioinformatics analysis, we found major subgroups of daptide BGCs that encode YcaOs, thiopeptide-like pyridine synthases, radical SAM enzymes, lanthipeptide dehydratases, luciferase-like monooxygenases, glycosyltransferases, and/or nucleotidyltransferases. These secondary PTMs of daptides are expected to confer structural and functional diversity. The bioinformatics survey of the daptide class also demonstrated different paradigms between Actinomycetota and Bacillota BGCs. Functionally equivalent daptide biosynthetic enzymes from different phyla are not phylogenetically similar, despite all surveyed daptide BGCs containing an NAD(P)-dependent oxidative decarboxylase, aminotransferase, and an RRE-containing DUF (now known to be critical for oxidative decarboxylation). The evolutionary distance between the daptide BGCs suggests that the pathways emerged individually in each phylum, and this may further divide the ecological role of daptides by phylum in addition to secondary modifications.

Peptidic natural products with amino-modified C-termini are rare, and the corresponding biosynthetic machinery remains elusive. Recently, the non-ribosomal aquimarins, which feature an amino-modified C-terminus, were isolated from sponge-derived bacterium *Aquimarina* sp. Aq135[43]. The amino-modified C-terminus was proposed to be generated by thioester reductase-catalyzed release of an aldehyde and subsequent transamination by an aminotransferase. In contrast, our work unveils a strategy for the amino-modification of ribosomal peptide C-termini. MpaB and MpaC collaboratively catalyze the oxidative decarboxylation of a C-terminal Thr into a ketone with concomitant loss of $CO_2$. PLP-dependent transamination and SAM-dependent N,N-dimethylation by MpaD and MpaM, respectively, yield the terminal (S)-$N_2$,$N_2$-dimethyl-1,2-propanediamine. The leader peptide is then removed by an RRE-dependent intramembrane peptidase (MpaP), yielding the final peptide that has converted a negatively charged C-terminal carboxylate into a positively charged tertiary amine.

Within RiPP biosynthesis, there are many other examples of C-terminal modifications. Of particular relevance are the oxidative decarboxylation reactions observed in the biosynthesis of mycofactocin (radical SAM), bottromycin (cytochrome P450), micrococcin/thiocillin (Bacillota DapC homolog), and Avi(Me)Cys formation (flavin-dependent decarboxylase)[2]. C-terminal modifications extend to peptide hormones (e.g., oxytocin and vasopressin) and antimicrobial peptides, which display decreased function without C-terminal amidation[44,45]. Additionally, GTPases employ a C-terminal modification strategy, including esterification of the C-terminus, to enhance insertion into cell membranes[46]. Nature has evolved numerous ways to neutralize local charge at the C-terminus through PTMs, but daptide biosynthesis goes one step further: generating a positive charge at the terminus. The biophysical implications of peptides with two positively charged amino-termini are not yet understood, but the combined hydrophobicity, helicity, and charge modification of the terminus suggest daptides may share a common membrane-targeting mode of action. Further studies of the substrate scope for Dmp formation will determine whether this charge swapping strategy will be useful for engineering peptides that associate with or penetrate membranes.

## Methods

### Bioinformatic discovery of the daptides

**Initial Analysis by RODEO.** All RiPP Recognition Element (RRE)-containing Uniprot identifiers (IDs) that were previously gathered using the exploratory mode of RRE-Finder[23] were converted to GenBank coding sequence (CDS) accession IDs using the Uniprot Retrieve/ID mapping tool. To identify potential precursor peptides in intergenic open reading frames (ORFs), all GenBank IDs were subjected to the genome mining tool Rapid ORF Description and Evaluation Online (RODEO)[25]. As part of standard RODEO analysis, '.gbk' files for each provided accession ID are downloaded from the National Center for Biotechnology Information (NCBI). From these '.gbk' files, nucleotide sequences were subjected to CDS prediction using Prodigal-shorter to identify ORFs. Prodigal-shorter is an adaptation of Prodigal-short used in the RiPPER workflow[47] that allows for CDSs to be predicted as short as five amino acids in length, based on recent reports of RiPP precursor peptides of that length[21]. CDS prediction of the RODEO selected region was performed in Prodigal's meta mode for records with fewer than 100,000 nucleotides. For larger records, a CDS training step was performed on the whole record before CDS prediction on the RODEO selected region. Intergenic ORFs were dereplicated by highest Prodigal score. These Prodigal scoring results were output to a separate '.csv' file, and a FASTA file was generated containing all genes and all possible predicted ORFs that were smaller than 120 amino acids and within 8 genes of an RRE domain.

**All-by-all BLAST analysis.** The list of GenBank CDS IDs was mapped to protein sequences in FASTA format using epost and efetch commands from the NCBI Entrez E-utilities. The FASTA file was converted to a blastdb using DIAMOND makedb[48]. An all-by-all BLAST search was performed using DIAMOND blastp on very-sensitive mode, a faster version of NCBI's Basic Local Alignment Search Tool (BLAST). The list of all ORFs was checked against the query list of RRE-containing proteins to remove any RRE domains from precursor peptide analysis.

Each ORF was also compared to a set of pHMMs adapted from RRE-Finder's precision mode, and all hits were removed. These ORFs were then converted to a blastdb, and an all-by-all BLAST was performed using DIAMOND.

**Prediction of paired RRE families and precursor families.** All-by-all BLAST results for the set of all RRE domains were parsed and entered into the precursor prediction algorithm (Fig. S1, Note S1). All RRE pairwise BLAST results with a bitscore <50 were removed. ORF all-by-all BLAST results were parsed, and pairwise BLAST results with bitscore <10 were removed. All self-hits were also removed.

All RRE domains were sorted into families using the presence or absence of pairwise BLAST results above the bitscore threshold (starting with 50). Families were sorted by descending order of length. Any RRE families with fewer than 30 members were omitted for simplified analysis of only families with high membership in the dataset. To maximize potential leads, cases where proteins in the RRE family were highly divergent were re-entered into the algorithm at a higher bitscore, increasing the threshold bitscore by 10 with each iteration. RRE divergence was assessed by "enzyme connectivity", which is the ratio of observed pairwise hits in an RRE family above the threshold bitscore to the maximum possible number of pairwise hits for the RRE family. For RRE families with enzyme connectivity ratios >0.2 and sufficient size, the size of the largest precursor family was calculated.

The list of ORF pairwise BLAST results was checked for results where both ORFs are encoded near RREs from the same family. Precursor families were generated from each list of related ORFs encoded near members of the same RRE family as above. The size of the precursor family was then compared to the size of the RRE family. When the size of the largest precursor family was within an order of magnitude of the size of the RRE family, the RRE family was output from the algorithm for manual evaluation. For other cases, the RRE family was re-entered into the algorithm recursively, with the bitscore threshold increased by 10. This recursive process resulted in the fractionation of RRE families, with each resultant family re-entered into the algorithm at the more stringent similarity threshold. Output paired families of RRE domains and precursor peptides were assessed based on enzyme connectivity, the relative size of precursor families, enzyme co-occurrence, and precursor peptide sequences for novelty and prioritization.

**Generation of pHMMs for daptide precursor peptides and RRE domains**

Precursor peptides output from the analysis algorithm were manually analyzed within the daptide family. Within the precursor family, 184 ORFs displayed Thr as the C-terminal residue. The 184 ORFs were aligned by Multiple Alignment using Fast Fourier Transform (MAFFT)[49] under the G-INS-I method. This alignment was used as input for HMMER3 hmmbuild[50] to create the daptide precursor peptide pHMM, which was subsequently uploaded to Skylign[51] for visualization.

Representative BGCs from Actinomycetota and Bacillota were selected. The last 100 amino acids (encompassing the entire predicted RRE domain) from the DapB proteins in these BGCs were copied into separate FASTA files by phylum. These files were then aligned using MAFFT under the G-INS-I method. These alignments were used as input for HMMER3 hmmbuild to create the Actino_DapB_RRE and Bacill_DapB_RRE pHMMs. The same approach was applied for DapP proteins from the same representative BGCs, using each protein's first 100 amino acids. Alignment with MAFFT and pHMM generation with hmmbuild generated the Actino_DapP_RRE and Bacill_DapP_RRE pHMMs.

**Expansion of daptide BGC family**

MpaB was submitted to PSI-BLAST analysis to identify additional BGCs within the daptide class. The first iteration was restricted to 100 protein hits, with subsequent iterations allowing up to 1000 protein

results, using an e-value cutoff of 10. After retrieving 1000 sequences, the accession IDs were submitted to RODEO for analysis and compilation.

The daptide precursor peptide pHMM was uploaded to the hmmsearch webtool hosted at EMBL-EBI[52]. The search was performed against UniProtKB, and the output results were analyzed for taxonomy. Individual IDs were converted to GenBank IDs using UniProt Retrieve/ID Mapping and analyzed by RODEO to confirm membership in the daptide class using local gene conservation. After identifying BGCs from Bacillota, an example DapB protein from *Bacillus cereus* (Genbank ID: KZD28690.1 [https://www.ncbi.nlm.nih.gov/protein/1017038289]) was selected for PSI-BLAST analysis using the method described for MpaB. These BGCs were analyzed and compiled with the results from the MpaB PSI-BLAST analysis.

Representative aminotransferase proteins in previously identified daptide BGCs were subjected to BLAST-P analysis on the NCBI's BLAST website. The top 2000 results were gathered for each aminotransferase, and duplicate results were removed. All resulting proteins were analyzed by RODEO. Putative BGCs were checked using local gene co-occurrence. BGCs containing a RODEO annotation for a DapB protein and either an iron-containing alcohol dehydrogenase (PF00465) or an enoyl-(acyl carrier protein) reductase (PF13561) were included as daptide BGCs, with all others removed. After using Prodigal dereplication strategies as described above, daptide precursor peptides were manually identified from ORF output to confirm membership in the daptide class.

**Generation of aminotransferase phylogenetic tree**

Outgroups for the phylogenetic tree were selected by analyzing the SwissProt database for class III aminotransferases. All daptide aminotransferases and three selected outgroups were aligned by MAFFT using the L-INS-I option. FastTree 2.1.10 was used to generate a maximum likelihood tree with the JTT + CAT model. The tree was visualized using the Interactive Tree of Life (iTOL) website (https://itol.embl.de/).

**Direct cloning of *mpa* and *sca* BGCs**

Direct cloning of *mpa* and *sca* BGCs was achieved by the CAPTURE method[28]. Both *M. paraoxydans* DSM 15019 and *S. capuensis* NRRL B-3501 were recovered on ISP2 agar medium (malt extract 10 g/L, yeast extract 4 g/L, glucose 4 g/L, agar 20 g/L, pH 7.2–7.4) at 37 °C until colony appears (about 3 days). A single colony was inoculated into 5 mL ISP2 liquid medium as seed culture and grown at 37 °C 250 rpm until saturation (3 d). 1 mL of the seed culture was then transferred into 50 mL fresh ISP2 liquid medium and cultivated for 18-20 h. Cells were then harvested by centrifugation at 3000 × g for 15 min and resuspended in 12 mL of cell resuspension buffer (50 mM Tris-HCl pH 8.0, 25 mM EDTA). Cell lysis enzymes (lysozyme 30 mg, mutanolysin 300 U), and 0.6 mg RNase A) were added and the sample was incubated at 37 °C for 18 h. Afterwards, 6 mg proteinase K was added and the sample was incubated at 37 °C for another hour. Following the incubation, 1.2 mL of 10% sodium dodecyl sulphate (SDS) was added and the sample was incubated at 50 °C for 2 h. Genomic DNA was recovered by phenol-chloroform extraction. The cell lysate was gently mixed with 15 mL phenol-chloroform-isoamyl alcohol (25:24:1, v/v, pH 8.0) until the aqueous phase became completely white. The sample was then centrifuged at 22,000 × g for 45 min at room temperature, and the aqueous phase was mixed with 10 mL chloroform. Following the same mixing procedure as described above, the sample was centrifuged at 22,000 × g for 10 min and the aqueous phase was aliquoted into 1.7 mL centrifuge tubes. DNA was recovered using isopropanol and sodium acetate precipitation, washed 3 times with 70% ethanol (v/v), and rehydrated in 10 mM Tris-HCl pH 8.0.

Guide RNA was prepared through in vitro transcription. Briefly, 5 μM forward and reverse template oligonucleotides were mixed in NEBuffer 3.1. The mixture was incubated at 98 °C for 5 min, followed by

slowly reducing the temperature with the rate of 0.1 °C/s until 10 °C was reached. A 2 μL aliquot of the annealed oligonucleotides was used as template for in vitro RNA transcription by HiScribe T7 quick high yield RNA Synthesis kit (New England Biolabs, MA). The transcribed RNA was purified using RNA clean and concentrator kit (Zymo Research, CA). Sequences of the DNA templates for the in vitro gRNA transcription were shown in Supplementary Table 1 and synthesized by Integrated DNA Technologies (Coralville, IA).

Genomic DNA was digested by FnCas12a in a 300 μL reaction containing 15 μg purified genomic DNA, 2.1 μg of each guide RNA, 60 pmol FnCas12a, and 30 μL of 10× NEBuffer 3.1. The reaction was performed at 37 °C for 2 h followed by 65 °C for 30 min. RNase A was then added to a final concentration of 0.1 mg/mL and the sample was incubated at 37 °C for another 30 min. Following RNase treatment, 3 μL of 20 mg/mL proteinase K solution was added and the sample was incubated at 50 °C for 30 min. The reaction was then transferred into 5PRIME PLG light phase-lock gel tubes (Quantbio, MA) and mixed with 300 μL of phenol-chloroform-isoamyl alcohol (25:24:1, v/v, pH 8.0). After proteins precipitated in the aqueous phase, the sample was centrifuged at 20,000 × g for 30 s. The aqueous phase was transferred into a new phase-lock tube and the extraction step was repeated. Ethanol and sodium acetate precipitation was then used to recover the DNA in the aqueous phase. The DNA pellets were washed 2 times with 70% ethanol (v/v) and rehydrated in 15 μL of 10 mM Tris-HCl pH 8.0.

The digested genomic DNA was assembled with DNA receivers by the T4 DNA polymerase exo + fill-in method. DNA receivers were PCR amplified using primers listed in Supplementary Table 1. A 15 μL assembly reaction consists of 3–3.75 μg of digested genomic DNA, 15 ng of DNA receiver amplified from plasmid pBE44, 35 ng of DNA receiver amplified from plasmid pBE45, 1.5 μL of NEBuffer 2.1, and 0.75 U of T4 DNA polymerase. Before adding T4 DNA polymerase, the sample was first incubated at 65 °C for 10 min followed by 25 °C hold without any mixing. T4 DNA polymerase was then added and gently mixed using wide-bore pipette tips. The reaction was performed at 25 °C for 1 h, 75 °C for 20 min, and 50 °C for 30 min. Next, 1 μL of 1 mM NAD⁺, 0.4 μL of 10 mM dNTPs, 1 μL (3 U) of T4 DNA polymerase, and 1 μL of E. coli DNA ligase were added to the reaction, which was incubated at 37 °C for 1 h, 75 °C for 20 min, and stored at 10 °C until transformation.

The assembled DNA fragments were then circularized by Cre-lox in vivo recombination. E. coli NEB10β cells containing the pBE14 helper plasmid were grown overnight at 30 °C in modified SOB medium (20 g/L Bacto tryptone, 5 g/L Bacto yeast extract, 10 mM NaCl, 2.5 mM KCl) supplemented with 8 μg/mL tetracycline hydrochloride. A 100 μL aliquot of the overnight culture was used to inoculate 10 mL modified SOB medium supplemented with 8 μg/mL tetracycline hydrochloride and cultured at 30 °C. After 2 h growth (OD600 of -0.2), 100 μL of 1 M L-arabinose was added to the culture and cells continued to grow at 30 °C until the OD600 reached 0.45-0.55 (-1.5 h after induction). Cells were then harvested by centrifugation at 3220 × g for 7 min at room temperature and washed using 1 mL of 10% (v/v) glycerol for three times. Cells were then resuspended in a final volume of 70 μL 10% (v/v) glycerol and gently mixed with 2.5 μL assembly reaction using wide-bore pipette tips. Electroporation was performed in 1 mm cuvettes using Gene Pulser XCell Electroporation system (Bio-Rad, CA) set at 1250 V, 100 Ω, 25 μF. Afterwards, cells were resuspended by 1 mL LB medium supplemented with 5 mM MgCl₂ and recovered at 37 °C for 75 min with shaking at 250 rpm. All the cells were plated on LB agar plates containing 50 μg/mL apramycin with blue/white screening and incubated at 37 °C until colonies appeared. White colonies were picked and grown overnight in 5 mL LB medium supplemented with 50 μg/mL apramycin at 37 °C. The plasmid DNA was purified from the cultures using Qiaprep Spin Miniprep Kit (Qiagen, Germany), digested by appropriate restriction enzymes, and analyzed by agarose gel electrophoresis. Correct plasmids were transformed into WM6026 (supplemented with 2,6-diaminopimelic acid at the final concentration 40 μg/mL for growth) and conjugated into S. lividans TK24 and S. albus J1074 for expression[53].

## Product analysis from colony extracts

The exconjugants of S. lividans TK24 and S. albus J1074 containing mpa, sca, or the empty pBE45 vector were picked and restreaked onto fresh MS (mannitol 20 g/L, soybean flour 20 g/L, agar 20 g/L) and ISP4 (soluble starch 10 g/L, K₂HPO₄ 1 g/L, MgSO₄·7H₂O 1 g/L, NaCl 1 g/L, (NH₄)₂SO₄ 2 g/L, CaCO₃ 2 g/L, FeSO₄ 1 mg/L, MnCl₂ 1 mg/L, ZnSO₄ 1 mg/L, agar 20 g/L) agar medium supplied with apramycin at a final concentration of 50 μg/mL and incubated under 30 °C for 5 d. A portion of cell mass (pinhead-sized) was picked from the plate, placed in 20 μL methanol and incubated at room temperature for 1 h. Methanol extract was then mixed equally with 1 μL of 15 mg/mL 70% aq. MeCN solution of α-cyano-4-hydroxycinnamic acid (CHCA) with 0.1% trifluoroacetic acid (TFA) (v/v) on a ground steel MALDI target, and the droplet was dried under ambient conditions. Samples were analyzed using a Bruker UltrafleXtreme MALDI-TOF MS using manufacturer methods for reflector positive mode. The MALDI LIFT-TOF/TOF mass spectra were acquired in the positive ion mode. Metastable fragmentation was induced by a nitrogen laser (337 nm) without the further use of collision gas. Precursor ions were accelerated to 8 kV and selected in a timed ion gate. In the LIFT-cell the fragments were further accelerated to 19 kV. The reflector potential was 29 kV.

## Heterologous expression and product isolation

Freshly obtained exconjugants of S. albus J1074 containing mpa or sca were individually restreaked onto MS plates with apramycin and incubated under 30 °C for 5 d. Colonies were verified for producing the daptides by MALDI-TOF MS by the method described above. The colonies were then scratched individually from the plate by sterile cotton swabs, spread on fresh MS medium with apramycin, and allowed to grow for another 5 d. The spores obtained from a single 150 × 20 mm plate were then transferred into 1 mL sterile water. An 100 μL aliquot of the spore solution was used to inoculate 50 mL bottromycin production medium (BPM; 10 g/L glucose, 15 g/L soluble starch, 5 g/L yeast extract, 10 g/L soy flour, 5 g/L NaCl, 3 g/L CaCO₃) supplied with 50 μg/mL apramycin in a 250 mL flask, which was shaken at 250 rpm for 4 d at 30 °C. Each 50 mL seed culture was then transferred into 500 mL fresh BPM supplied with 50 μg/mL apramycin in a 2 L flask and shaken at 250 rpm for another 5 d at 30 °C.

The cells were then harvested by centrifugation and mixed with methanol (1/10 volume of the liquid culture) to homogeneity with continuous stirring under room temperature for at least 1 h. Cell debris was then removed by centrifugation, and the methanol extract was mixed with an equal volume of water and loaded onto an Agilent Bond Elut C18 Solid Phase Extraction (SPE) column (bed mass, 10 g; volume, 60 mL; particle size 120 μm), which was pre-equilibrated by 50 mL 5% B (solvent A = 0.1% TFA in water; solvent B = 0.1% TFA in acetonitrile). The compounds were then eluted using a step gradient with increasing percentage of solvent B in 150 mL volumes: 5%, 20%, 30%, 40%, 50%, 60%, 70%, 80%, and 100% B. The eluted compounds were monitored by MALDI-TOF MS. Fractions containing the compounds (50%, 60%, and 70% B) were lyophilized to dryness and powders were redissolved into methanol. Semi-reparative HPLC purification was performed using a 1290 Infinity II Preparative LC System equipped with a Phenomenex Luna C5 column (5 μm, 100 Å, 250 × 10 mm) equilibrated in 5% B. Compounds were eluted by an increase to 100% B over 20 min with a flow rate of 3 mL/min. Under these conditions, daptides **1**, **2**, and **3** were eluted at 16.7, 15.5, and 14.4 min, respectively. All fractions were analyzed by MALDI-TOF MS, lyophilized to dryness, and stored at -80 °C until further use. Typical yields were 0.2 – 0.3 mg per liter of BPM.

## Antimicrobial activity assay

Daptides were dissolved in DMSO to achieve a concentration of 10 μM. Agar plates were prepared by combining 20 mL of melted solid medium (cooled to 42 °C for 5 min) with 200 μL of stationary phase overnight cell culture. The seeded agar was poured into a sterile 100-mm round dish (VWR) and allowed to solidify at 25 °C for 10 min. Daptides were directly spotted on the solidified agar. Plates were incubated at various temperatures shown in Supplementary Table 4 for 16 h, and the antimicrobial activity was determined by the presence or absence of zones of growth inhibition.

## Hemolytic Assay

Fresh defibrinated whole bovine blood was obtained from Hemostat Laboratories. Whole blood was washed three times in PBS and diluted to a final concentration of 1:25 vol/vol in PBS. Prepared whole blood was then split into 50 μL aliquots in individual 1.7 mL Eppendorf Tubes. Next, stock solutions of concentrations at 1 mM and 200 μM were prepared for **1**, **2**, and **3** in DMSO, and **1** + **2**, **2** + **3**, **1** + **3** and **1** + **2** + **3** stock solutions at 1 mM and 200 μM were then prepared by mixing equal volumes of the single daptide stock solutions accordingly. Aliquots of 2.5 μL each stock solution were mixed with the blood, yielding final concentrations at 50 μM and 10 μM. An equal volume of DMSO and Triton X-100 were used as negative and positive controls. The mixtures were then incubated in Eppendorf Thermoxixer C with a heated lid for 18 h at 37 °C. After incubation, the samples were processed by centrifugation at 500 × g for 10 min, and the supernatants were measured for hemoglobin absorbance at 410 nm on a NanoDrop Spectrophotometer. Each measurement was performed in biological triplicate, and the percentage of hemolysis was calculated using Equation 1.

$$\%\text{Hemolysis} = \frac{\text{Abs of test sample} - \text{Abs of DMSO}}{\text{Abs of Triton X}-100 - \text{Abs of DMSO}}. \quad (1)$$

## Construction and characterization of the minimal *mpa* BGC in *S. albus* J1074

The complete or partial *mpa* operons were PCR amplified from the directly cloned *mpa* BGC as multiple DNA fragments. The primers were designed to generate an 80 bp overlapped region between adjacent DNA fragments (Supplementary Table 5). The *E. coli-Streptomyces* shuttle vector pSET152 was digested by *Xba*I and *Eco*RI. The *S. cerevisiae* helper fragment was amplified from the plasmid pRS416. All DNA fragments were gel-purified from 0.7% agarose. For each construct, 200 ng vector was mixed with other DNA fragments at a 1:1 molar ratio, concentrated by DNA Clean & Concentrator-5 (Zymo Research, CA) and eluted into 6 μL of water. The DNA mixture was then assembled by DNA assembler[36]. Single colonies of *S. cerevisiae* YSG50 were inoculated in 2 mL YPAD medium (yeast extract 10 g/L, peptone 20 g/L, glucose 20 g/L, adenine sulfate 40 mg/L) and grown overnight at 30 °C and 250 rpm. A 0.5 mL aliquot of the overnight culture was transferred into 50 mL fresh YPAD medium and shaken at 250 rpm and 30 °C for 4–5 h until OD 600 reached 0.8–1.0. Cells were harvested by centrifugation at 3000 × g for 5 min at 4 °C. The cell pellet was washed by 50 ml ice-cold water, followed by another wash by cold 1 M sorbitol, and resuspended in 250 μL cold 1 M sorbitol. An aliquot of 50 μl of yeast cells was mixed with 4 μL DNA mixture and electroporated in a 0.2 cm cuvette at 1.5 kV. Afterwards, the cells were resuspended by 1 mL room temperature YPAD medium and recovered at 30 °C with shaking at 250 rpm for 1 h. Cells were then harvested by centrifugation, washed by 1 mL of 1 M sorbitol at room temperature for two times, spread on Sc-Ura plates, and incubated at 30 °C for 2–4 days until colonies appeared. Single colonies were grown in Sc-Ura medium for 1 d and plasmid DNA was purified using Zymoprep II Yeast plasmid Miniprep kit (Zymo Research, CA). Plasmids were then transformed into *E. coli* NEB10β, purified by Qiaprep Spin Miniprep Kit (Qiagen, Germany), and analyzed by restriction digestion. Correct plasmids were conjugated into *S. albus* J1074, and colony extracts were analyzed by MALDI-TOF MS as described above.

## Refactoring and gene omission of the *mpa* pathway

The *mpaA1-3*, *mpaB*, *mpaC*, *mpaD*, and *mpaM* genes were codon-optimized and synthesized and then subcloned onto helper plasmids for a plug-and-play refactoring strategy (Supplementary Table 5, Supplementary Table 6)[37]. Point mutations to *mpaA1* were introduced by overlap extension PCR. The *mpaA1* and its mutants obtained by PCR were then cloned into T7 His Helper-1 for constructing plasmids with other genes. The refactored pathway was built by the Golden Gate assembly[37]. The 4 bp adapters affected by gene omission were changed by PCR with primers that anneal to the T7 promoter and terminator region. The resulting PCR fragment was used as the insert in the Golden Gate assembly.

## Expression of the refactored *mpa* pathways and product purification

Unless otherwise specified, the following conditions were used for the co-expression of MpaA1 and variants thereof with other biosynthetic genes in *E. coli*. The plasmids containing refactored *mpa* pathways were co-transformed with the chaperone plasmid (pGro7) into *E. coli* BL21(DE3). Cells were grown on Luria–Bertani (LB) agar plates containing 50 μg/mL kanamycin and 20 μg/mL chloramphenicol at 37 °C overnight. Single colonies were picked to inoculate 5 mL LB supplied with the same amount of antibiotics and grown at 37 °C for another 16 h. This culture was used to inoculate 500 mL M9 medium supplied with 50 μg/mL kanamycin, 20 μg/mL chloramphenicol, and 0.5 mg/mL arabinose. The M9 culture was grown at 37 °C to an optical density of 600 nm (OD600) of 0.6. Isopropyl β-d-1-thiogalactopyranoside (IPTG) was then added to a final concentration of 0.5 mM. The cell culture was then cooled to 18 °C and grown for an additional 3 d.

Cells were then harvested by centrifugation and resuspended in lysis buffer (6 M guanidine hydrochloride, 20 mM NaH$_2$PO$_4$, 500 mM NaCl, 0.5 mM imidazole, pH 7.5) to a final volume of 20 mL and lysed by sonication. The cell lysate was then clarified by centrifugation at 23,700 × g for 30 min at 4 °C, and the supernatant was passed through a syringe filter (0.45 μm). The clarified lysates were loaded onto a 5 mL NiNTA HisTrap column (GE Healthcare). The column was then washed with 25 mL of wash buffer (4 M guanidine hydrochloride, 20 mM NaH$_2$PO$_4$, 500 mM NaCl, 30 mM imidazole, pH 7.5) and eluted with 15 mL of elution buffer (4 M guanidine hydrochloride, 20 mM Tris, 100 mM NaCl, 1 M imidazole, pH 7.5). The eluent was desalted by ZipTip or Bond Elut C18 SPE column and analyzed by MALDI-TOF MS.

## Derivatization of ketone intermediate

A refactored plasmid containing genes *mpaA1*, *mpaB*, and *mpaC* was heterologously expressed as described above. After desalting by SPE column, 1 mL (sample in aq. 70% acetonitrile/0.1% TFA) was added each to two scintillation vials (reaction vs. control). The pH of the eluent was adjusted to pH 4 using 0.1 M NaOH and checked by pH paper. To one vial, *O*-benzylhydroxylamine was added to 10 mM and scintillation vials were left overnight (~16 h) at room temperature to ensure maximal oxime formation. Reaction products were mixed 1:1 with 50 mg/mL Super-DHB (Sigma-Aldrich) in aq. 60% acetonitrile/0.1% formic acid and dried under ambient conditions on a polished steel MALDI target. Samples were analyzed using a Bruker UltrafleXtreme MALDI-TOF MS using manufacturer's methods for reflector positive mode.

## AlphaFold-Multimer Structural Prediction of MpaA1-MpaB-MpaC Complex

AlphaFold-Multimer was used to predict the MpaB-MpaC-MpaA1 interactions, with each of the five trained model parameters[40]. The

MSA generation, AlphaFold-Multimer predictions, and structure relaxation with Amber were run using the code of ColabFold, a publicly available Jupyter notebook[54], on a Google Colab GPU cluster. The input included the query sequences of MpaB, MpaC, and MpaA1, with an MSA from MMseqs2 (UniRef+Environmental) not using any templates. The Pair mode "unpaired+paired" and the number of recycles, "6," were selected in the advanced settings section. The structure of MpaC with ligand NADP was constructed using a homologous alcohol dehydrogenase (PDB ID: 6C76) as a template followed by energy minimization using software YASARA Structure version 17.8.19[55–57]. The protein-protein interaction was visualized and analyzed using PyMOL version 2.5.4[58] and Chimera version 1.16[59].

### Reporting summary
Further information on research design is available in the Nature Portfolio Reporting Summary linked to this article.

## Data availability
We declare that all data supporting the findings of this study are presented in the main manuscript text, its Supplementary Information and Supplementary Data files. NCBI accessions used in this paper were obtained from NCBI (https://www.ncbi.nlm.nih.gov/) and include KZD28690.1 [https://www.ncbi.nlm.nih.gov/protein/1017038289]. PDB accessions were obtained from RCSB PDB (https://www.rcsb.org/) and include 6C76. Additional NCBI, PDB, and Uniprot (https://www.uniprot.org/) accessions are referenced in the Supplementary Information and Supplementary Data, and these accessions are publicly accessible on the respective NCBI, RCSB PDB, and Uniprot websites. Source data are provided with this paper. Data is available from the corresponding authors upon request. Source data are provided with this paper.

## Code availability
All code and methods necessary to reproduce this work, including all custom pHMMs, are available in the Supplementary Information file and on Github (https://github.com/the-mitchell-lab/rodeo2).

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

## Acknowledgements

We wish to thank Sangeetha Ramesh and Timothy Precord for HR-MS/MS assistance, Lonnie Harris for assistance with Marfey's method, Tom Pires for UHPLC method development, and David Slater for providing access to AlphaFold-Multimer. This work was supported in part by grants from the National Institutes of Health (GM123998 to D.A.M. and AI144967 to D.A.M. and H.Z.).

## Author contributions

H.R. and S.R.D. contributed equally to this work. H.R. performed direct cloning, heterologous expression, mutational analysis, compound purifications, and bioactivity assays. S.R.D. performed bioinformatics, HR-MS/MS analysis, and stereochemical and functional group determinations. Y.P. assisted in the preparation of 1 for NMR. C.H and L.Z. designed, performed, and analyzed NMR experiments. M.N. synthesized Dmp standards with oversight from D.S. S.R.D. and H.C. performed AlphaFold-Multimer analysis. H.R. and S.R.D. wrote the manuscript with editorial oversight from D.A.M. and H.Z., along with input from all other authors. D.A.M. and H.Z. conceived of and supervised the overall project.

## Competing interests

The authors declare no competing interests.
