## [Peer Review File · Nature Communications]

Genome mining unveils a class of ribosomal peptides with two amino terminiREVIEWER COMMENTS

Reviewer #1 (Remarks to the Author):

Using a widespread specific Ripp domain called RiPP recognition element (RRE), Ren et al., identified a completely new family of Ripples called Daptides defined by an unusual (S)-2-(dimethylamino)propylamine (Dap)-modified C-terminus. The authors were able to isolate and elucidate the chemical structure of a representative compound and perform first biosynthetic studies. The manuscript includes a wealth of data, is well written, and of high impact to the field. I am very curious to see more about the exact bioactivity and function of these compounds in the future and have only some minor comments, which in my opinion would improve the manuscript.

- 1) The taxonomic distribution of BGCs of the identified daptipeptide BGCs is shown in a table in the supplements. However, maybe a figure visualizing specific distributions of the BGCs along with the "core" genes of the daptides and phylum specific differences in tailoring enzymes would provide a better overview of this new class of Ripples.
- 2) Line 315 to 318: could you provide any evidence that the BGCs in the different phyla emerged completely independent? This would be an interesting case of convergent evolution, which is not supported any data presented in the manuscript so far.
- 3) 496 typo: was used as... instead of "as used as..."

Reviewer #2 (Remarks to the Author):

The paper by co-workers from the Mitchell and Zhao labs is a complete piece of work, from bioinformatics prediction to genetic, biochemical and chemical analysis (biosynthetic logic and mode of action) of an unprecedented type of RiPP BGC overlooked until now. Through a combined computational and wet-lab approach authors characterize the emerging features of this novel class of RiPPs, which includes an oxidative mechanism that renders a dimethylamino propylamine terminus at the carboxyl terminal-end of the so-called daptides RiPPs. The relevance of this discovery is further expanded into a potential novel mode of action involving the positive charge of these peptides at their carboxyl-end. However, only one BGC / one metabolite was characterized and the claims are quite broad, possibly beyond what the data provided can support.

Major.

The (first) bioinformatics result's section is hard to read with plenty of general details presented in a generalized fashion. Indeed, Fig. 1 lacks these results and looks more as a graphical abstract / ppt slide. All the results are certainly included as supplementary information, somehow, but these are not so accesible, and are not cited within the legend to Fig. 1.

Please provide key results within Fig. 1 and not only cartoons.

The latter could include some of the supplementary Figures, but also some sort of BiG-SCAPE and/or CORASON analysis of the identified and newly predicted DAP BGC, clearly showing the diversity (if any) and location of the BGCs from Microbacterium/Staphylococcus, within others.

Line 171 "To evaluate this hypothesis, a sufficient amount of 1 was accumulated for structure determination, due to its relatively higher post-purification yield." This observations highlights the importance of the previously requested analysis (new Figure 1), as it raises the question about how conserved is this BGCs and their cognate products.

Being the claim that this is a novel class of BGC with very distinctive chemical features, the paper will enormously benefit from further examples being characterized (selected after well-defined criteria,

based in the data analysis shown in Figure 1).

* If the authors feel that reporting on further BGCs is not viable, then please provide an honest justification... but tune down the conclusions in line with this major limitation.

Line 192 “Examination of daptide bioactivity” The first paragraph of this section reads more like a Discussion. Indeed, first sentence reads like a big analysis when it seems, at least to this reviewer, that a metabolite from the literature was noticed (I may be wrong, but this is how it reads).

From this observation, authors used this antecedent to direct their bioactivity experiments, but this can be said in one single line. Moreover, without the genome sequence of the hominycin producing organism the above mentioned request for more examples is further supported.

Could you actually obtain the genome sequence of this strain to make this observation a result? After all, it is a type DSM strain and genomic DNA can be purchased. The speculation of *Staphylococcus pseudintermedius* having a DAP-containing homologous daptide BGC is not sustained without further data, and even doubtful given the different bioactivity results (paper is from 2010 and may be good to revisit it in the light of these results).

In any case, it would be desirable to see this BGC included as part of the changes, information, requested above for Fig. 1.

* Could you mention hominycin in the Introduction as an example of RiPPs that lack genetic basis for their synthesis yet have interesting chemical modifications?

The *E. coli* heterologous expression experiments are as robust and clearly presented as they can possibly be. I just wish the authors can provide more examples given the synthetic / HTP platform they have used for this purpose, which they seem to master.

In the absence of this, it would be desirable to see a similar analysis but with mutants of the *Microbacterium* strain used, justifying the one-strain centric approach to call upon a complete new BGC class. At least the risk of having contrasting results between the heterologous and homologous systems could be ruled out.

The section “Oxidative Decarboxylation of the C-terminal Thr” heavily relies on the results (not shown) of Figure 1, justifying again the request for a comprehensive new Figure 1.

I feel the AlphaFold section, which includes again the superlative “impressive”, is anecdotal and adds little value to the main purpose of the paper: a new DAP BGC with novel chemical, genetic and mechanistic features that warrant further in-depth investigation (we are all impressed by AlphaFold, but this paper is not about the proteins themselves but how these are involved in an unprecedented biosynthetic logic – Figure 5 is not justified).

* Could AlphaFold analysis / depiction be somehow part of Fig. 1 together with the rest of the bioinformatics analysis? This could help to particularly highlight the occurrence of two RRE domains and their distribution. An alternative could be to include Fig. 5 in Figure 4, but not by itself.

Minor.

Abstract. Please include the name of the *Microbacterium* strain used for validation of the new DAP class.

Line 61... “Impressive new RiPP classes...” avoid superlative adjectives unless it is explained in which way these metabolites are “impressive”

Line 67... “are also involved in primary metabolism”. This statement is misleading in the way that

primary metabolic enzymes can be a means to successfully generate genome mining predictions in a class-independent fashion. See EvoMining by Barona and co-workers. Not sure if EvoMining has been used for RiPP purposes but this needs to be clarified.

Line 150... "After dataset analysis..." Please clearly state in which way and what were the considerations for this selection, analysis.

Line 153... "To validate the bioinformatics study..." If I am right, authors validate one single prediction and the entirety of the bioinformatics pipeline, analysis. Please change this.

Line 255. "The gene deletion studies showed..." Please modify this to avoid confusion with real homologous gene deletion experiments. The *E. coli* experiments are combinatorial heterologous expression or BGC reconstitution experiments, which can lead to results that may differ with homologous systems.

Line 302 – 318. This is the right place to discuss the homininin observation, but I was wondering if a comparison between Gram positives and Gram negatives would also be interesting. This could be done both in terms of the distribution of the BGC (again Figure 1, complete) and/or the membrane-related MoA postulated.

Reviewer #3 (Remarks to the Author):

Ren and colleagues report the discovery of the daptides, an unusual class of Ribosomally and Post-translationally modified Peptides (RiPPs) characterized by a novel C-terminal modified dimethylaminopropylamine (Dap). The authors pursued a unique in silico rationale for discovering new RiPP classes. Where previous studies mainly looked for biosynthetic enzyme homologs, the present study uses an alternative strategy searching for conserved recognition motifs for binding the RiPP's prepeptide leader sequence. Using their previously developed PRE-Finder algorithm, the authors assigned an impressive ~500 biosynthetic gene clusters (BGCs) to the daptide family, making this one of more abundant RiPP classes to date. The authors proceeded to characterize the BGC from *Microbacterium paraoxydans* as a type compound for the family. Using a refactoring approach, the BGC was heterologously expressed, leading to the identification of daptides 1-3, which contain the C-terminal Dap residue. Combining genetic engineered BGC variants and mass-spec analysis, the three-step conversion of the C-terminal threonine in the propeptide to Dap was elucidated. Because of the dual charged peptide ends, the authors hypothesized a bioactive role for the daptides in interacting with the membrane and the compounds were shown to demonstrate hemolytic activity. I really enjoyed the paper, and have no further suggestions for its improvement. The experimental procedures seem solid and I think this is an important and exciting new discovery that would be of broad interest in the (RiPP) natural products field. Given the scarcity of enzymes to modify C-termini, this work holds promise for future applications peptide engineering.

Reviewer #4 (Remarks to the Author):

In this paper, the authors report the discovery of a new class of RiPPs by using a RRE-based genome mining approach. This strategy relies on previous bioinformatics analysis tools such as RRE-Finder and RODEO.

The results are overall comprehensive, well supported and very well presented. This work embodies an advance in the class-independent genome mining strategies.

The main goal of the work is the application of a class-independent genome mining approach to successfully uncover a new RiPP class (named daptides) containing a characteristic 2-(dimethylamino)propylamine (abbreviated as Dap) modification at the former C-terminus. The authors showed this modification (the biosynthetic machinery responsible for that) is present in many other RiPP BGCs. The previously reported antimicrobial peptide hominigin also contains this type of modified C-terminus, therefore hominigin is retrospectively identified as a known member (the first discovered) of the same RiPP class.

The “Dap” biosynthetic pathway was identified by combining bioinformatic analysis and a refactoring strategy involving gene omission experiments. Although none of those enzymes were expressed and biochemically characterized, this is nonetheless a key result of the work and opens the way to engineer biosynthetic pathways with this type of C-terminal modification. The identification of residues within leader peptide involved in the interaction with RRE, and the AlphaFold Multimer-based modeling of the protein-protein interaction between the RRE, MpaB and MpaC are also interesting aspects of the work.

The main limitation of this approach is obvious: It is restricted to those RiPP classes containing recognition elements (RRE), i.e., about the half of prokaryotic RiPP classes, as the authors claim. Moreover, although the main objective is to find new RiPP classes through genome mining, -thereby avoiding some intrinsic problems of the screening-based approaches-, the fact is that such C-terminal modification is not really new, as it had already been uncovered with hominigin, which was discovered through bioactivity-guided screening.

To better understand the scope of this class-independent genome mining approach, it would be nice that authors show other RRE families found as output of this mining.

A few considerations before publication:

1. I think authors should include in the manuscript or in the supplementary material some data on the number and types of different “RRE families” found with this approach.

2. Given the Dap-C terminus modification is present in many distantly related BGCs, it appears to be a widespread PTM that may have emerged individually in each phylum (as the authors effectively say), rather than a class-defining feature in the strict sense.

The authors found that all those BGCs contain an invariant Thr residue at the C-terminus. However, I could not find the corresponding “T” hallmark in the Skylign logo (supplementary figure 2).

3. If only as a side comment, I think the abbreviation “Dap” used for naming the 2-(dimethylamino)propylamine moiety and the term “daptides” for the new class are confusing, since “Dap” is commonly used to refer to diaminopropionic acid or more rarely, diaminopimelic acid. The same moiety was abbreviated as “Dmp” in hominigin.

4. The authors acquired 2D NMR spectra of 1 but only show summarized data for the Dap moiety in supplementary Table 2. In that table, carbon chemical shift of C-3 methyl is missing (“not available”), but in the HSQC spectrum of 1 (supplementary figure 14), a cross-peak at ca. 6-6.5 ppm appears to correlate with the methyl protons signal at 1.34 ppm. Is this signal due to any impurity? Did the authors avoid this assignment (as a genuine signal in 1) for any reason?

In addition, the CH-2 proton signal is indicated as “br s” but should be noted as a multiplet (m), even although it was an apparent broad singlet.

5. NMR assignments of the Dap moiety (except that of C-3) are well supported and consistent with the structure proposed for daptide 1. The connectivity of Dap to the C-terminus of daptides was supported by HR-MS/MS in 2 and 3. However, the only spectroscopic evidence for connectivity between Dap and the rest of the peptide (1) is an HMBC correlation with an amide carbonyl at 175.8 ppm, which in turn was not assigned to any amino acid residue in the peptide.

I believe that full structural characterization (NMR assignments) of the compound should be done to

better support the structure of these new RiPPs. I think the authors could have put more effort on this issue. If overlaps in NMR spectra of 1 would make it impossible to complete, perhaps the assignments of peptides 2 or 3 (smaller and with fewer redundant residues) might be easier.

6. As for the Marfey's analysis of 1, the authors did not find the FDAA adducts of Trp and Tyr, even although it was hydrolyzed under relatively mild conditions (DCI, 95 C, 2 h; 3% phenol). Trp degradation is a common issue but that of Tyr is rarer under these conditions. Did the authors interrogate the LC-MS chromatograms for both mono- and di-FDAA derivatives of Tyr?

7. In Figure 3b, the MALDI-TOF MS analysis of the mpaABCDM full construct shows the presence of intermediates 5/6 and the final product 7, but also a compound at lower m/z (unlabeled in the figure). Any idea of the identity of this species?

8. "Expression of mpaABC (omission of mpaM and mpaD) resulted in the accumulation of the ketone 5, consistent with transaminase activity for MpaD". The omission of only mpaD should have led the same result. Why did the authors simultaneously omit both mpaM and mpaD?

Reviewer #1

The taxonomic distribution of BGCs of the identified daptipeptide BGCs is shown in a table in the supplements. However, maybe a figure visualizing specific distributions of the BGCs along with the “core” genes of the daptides and phylum specific differences in tailoring enzymes would provide a better overview of this new class of Ripps.

Response: We appreciate the suggestion from the reviewer. We have added a chart in Figure 1 showing the percentage of core and tailoring genes to each phylum. We direct interested parties to Supplementary Dataset 1, which contains all of the precise details regarding phylogenetic distribution and BGC composition.

Line 315 to 318: could you provide any evidence that the BGCs in the different phyla emerged completely independent? This would be an interesting case of convergent evolution, which is not supported any data presented in the manuscript so far.

Response: We agree that the convergent evolution within the daptides would be interesting. Unfortunately, the data we have is not conclusive; however, it does suggest convergent evolution by the substitution of functionally homologous protein families, *i.e.* Fe-dependent ADH and short chain oxidoreductase, across the two phyla as shown in the revised Fig. 1. Phylogenetic analysis of the biosynthetic proteins was difficult due to the lack of sequence similarity between the DUF-RRE proteins, alcohol dehydrogenases, methyltransferase, and peptidases. Additionally, the aminotransferase phylogenetic tree cannot definitively determine convergent evolution either. Providing more direct evidence of this comparison may be a point of further investigation when further BGCs are characterized.

496 typo: was used as... instead of "as used as..."

Response: Thanks a lot for the suggestion and the typo has been fixed.

Reviewer #2

The paper by co-workers from the Mitchell and Zhao labs is a complete piece of work, from bioinformatics prediction to genetic, biochemical and chemical analysis (biosynthetic logic and mode of action) of an unprecedented type of RiPP BGC overlooked until now. Through a combined computational and wet-lab approach authors characterize the emerging features of this novel class of RiPPs, which includes an oxidative mechanism that renders a dimethylamino propylamine terminus at the carboxyl terminal-end of the so-called daptides RiPPs. The relevance of this discovery is further expanded into a potential novel mode of action involving the positive charge of these peptides at their carboxyl-end. However, only one BGC / one metabolite was characterized and the claims are quite broad, possibly beyond what the data provided can support.

Major.

The (first) bioinformatics result's section is hard to read with plenty of general details presented in a generalized fashion. Indeed, Fig. 1 lacks these results and looks more as a graphical abstract / ppt slide. All the results are certainly included as supplementary information, somehow, but these are not so accessible, and are not cited within the legend to Fig. 1.

Please provide key results within Fig. 1 and not only cartoons.

The latter could include some of the supplementary Figures, but also some sort of BiG-SCAPE and/or CORASON analysis of the identified and newly predicted DAP BGC, clearly showing the diversity (if any) and location of the BGCs from Microbacterium/Staphylococcus, within others.

Response: We thank the reviewer for the suggestion. To address this critique, we have added two additional panels to Fig. 1 that show the precursor peptide logo and a direct comparison of pHMM frequencies between BGCs identified from the two major daptide-producing phyla to clearly depict the bioinformatics results. Additionally, the explanation of the bioinformatics workflow in Supplementary Fig. 1 has been elaborated to include a visual depiction of the logical basis for the algorithm.

Line 171 “To evaluate this hypothesis, a sufficient amount of 1 was accumulated for structure determination, due to its relatively higher post-purification yield.” This observation highlights the importance of the previously requested analysis (new Figure 1), as it raises the question about how conserved is this BGCs and their cognate products.

Response: In addition to changes in Fig. 1 to show the representativeness of this BGC, line 171 has been edited to clarify the statement and now reads “due to its higher post-purification yield than 2 or 3.” We feel that this statement will more clearly emphasize the intra-BGC comparison of peptide yields.

Being the claim that this is a novel class of BGC with very distinctive chemical features, the paper will enormously benefit from further examples being characterized (selected after well-defined criteria, based in the data analysis shown in Figure 1).

* If the authors feel that reporting on further BGCs is not viable, then please provide an honest justification... but tune down the conclusions in line with this major limitation.

Response: We appreciate the suggestion from the reviewer. Given the bioinformatics analysis, structural characterization, and biosynthetic investigation in this paper, we feel that our description of the daptides as a new compound class is within reason, despite the original report of only one BGC. That being said, we have cloned and heterologously expressed another representative BGC identified from *Streptomyces capuensis* NRRL B-3501 to further support the claim that daptides constitute a novel class of RiPPs. The products were analyzed by MALDI-TOF-MS, MALDI LIFT-TOF/TOF MS, and Marfey’s chiral analysis. Together, these indicate that the new products contain Dmp-modified C-termini, like daptides 1-3, and D-alanine residues, known chemistry for the added tailoring enzymes.

Line 192 “Examination of daptide bioactivity” The first paragraph of this section reads more like a Discussion. Indeed, first sentence reads like a big analysis when it seems, at least to this reviewer, that a metabolite from the literature was noticed (I may be wrong, but this is how it reads).

From this observation, authors used this antecedent to direct their bioactivity experiments, but this can be said in one single line.

Response: We appreciate the suggestion from the reviewer. We agree that the first paragraph in “Examination of daptide bioactivity” is more appropriate for the Discussion session. We have moved the discussion of hominycin and the identification of its potential BGC to the Discussion session and shortened the rationale that guided the daptide bioactivity experiments.

Moreover, without the genome sequence of the hominycin producing organism the above mentioned request for more examples is further supported.

Could you actually obtain the genome sequence of this strain to make this observation a result? After all, it is a type DSM strain and genomic DNA can be purchased. The speculation of *Staphylococcus pseudintermedius* having a DAP-containing homologous daptide BGC is not sustained without further data, and even doubtful given the different bioactivity results (paper is from 2010 and may be good to revisit it in the light of these results).

In any case, it would be desirable to see this BGC included as part of the changes, information, requested above for Fig. 1.

Response: We appreciate the suggestion from the reviewer. However, the *Staphylococcus pseudintermedius* strains (B32 and isolate 17S01341-1) that were predicted to encode daptide BGCs are not available in the DSMZ or any other publicly available strain collections. In addition, the hominycin producing strain *Staphylococcus hominis* MBBL 2-9 was also not publicly accessible. Instead of further pursuing the BGCs that may produce hominycin-like products, we decided to use the relevant BGC from *Streptomyces capuensis* as another example to support our claim of daptides as a new class of RiPPs.

* Could you mention hominycin in the Introduction as an example of RiPPs that lack genetic basis for their synthesis yet have interesting chemical modifications?

Response: We appreciate the suggestion from the reviewer. However, we believe that discussion of hominycin as an example of RiPPs that lack genetic basis in the Introduction may not be appropriate. Discussing hominycin in the Introduction might draw confusion, as there are more well-known cases of reported compounds that were later discovered to be RiPPs, e.g. polytheonamides. Also, peptidic natural products can be synthesized via ribosomal or nonribosomal synthesis, so hominycin could not definitively be claimed as a RiPP in the previous paper, and its biosynthesis was not previously discussed.

The *E. coli* heterologous expression experiments are as robust and clearly presented as they can possibly be. I just wish the authors can provide more examples given the synthetic / HTP platform they have used for this purpose, which they seem to master.

In the absence of this, it would be desirable to see a similar analysis but with mutants of the *Microbacterium* strain used, justifying the one-strain centric approach to call upon a complete new BGC class. At least the risk of having contrasting results between the heterologous and homologous systems could be ruled out.

Response: We appreciate the suggestion from the reviewer. As mentioned above, we have also selected a relevant BGC from *Streptomyces capuensis* as another example for characterization. The *Microbacterium* strain was cultured in multiple media and solvent-extractable metabolites were meticulously analyzed; we conclude that the BGC is silent under the culturing conditions we screened. Genetic tools in *Microbacterium* strains are also not viable, making heterologous expression the optimal method for investigation.

The section “Oxidative Decarboxylation of the C-terminal Thr” heavily relies on the results (not shown) of Figure 1, justifying again the request for a comprehensive new Figure 1.

I feel the AlphaFold section, which includes again the superlative “impressive”, is anecdotal and adds little value to the main purpose of the paper: a new DAP BGC with novel chemical, genetic and mechanistic features that warrant further in-depth investigation (we are all impressed by AlphaFold, but this paper is not about the proteins themselves but how these are involved in an unprecedented biosynthetic logic – Figure 5 is not justified). These models are actually relevant to the bioinformatic analysis and illustrate the pipeline

* Could AlphaFold analysis / depiction be somehow part of Fig. 1 together with the rest of the bioinformatics analysis? This could help to particularly highlight the occurrence of two RRE domains and their distribution. An alternative could be to include Fig. 5 in Figure 4, but not by itself.

Response: We thank the reviewer for their suggestions. We have re-worked Fig. 1 to include more direct results. The word “impressive” has been removed. While we agree with the reviewer that the main focus of the paper is not on the proteins, MpaB lacked any previous annotation, and it was the RRE domains that permitted discovery of the new class. As a result, we elected to provide more supportive data (rather than less data) to allow other groups to follow a parallel approach. Inclusion of these predictive models also suggested a putative function to the DUF, and without it, we would have an empty spot in the proposed biosynthetic pathway. Nevertheless, we agree that the AlphaFold models are not the centerpiece of this manuscript, so we have combined Figs. 4 and 5 to show the oxidative decarboxylation of Thr.

Minor.

Abstract. Please include the name of the *Microbacterium* strain used for validation of the new DAP class.

Response: We thank the reviewer for their suggestions. The name of the strain has been added to the abstract and the first mention of the strain in the text.

Line 61... “Impressive new RiPP classes...” avoid superlative adjectives unless it is explained in which way these metabolites are “impressive”

Response: We thank the reviewer for the suggestion. The word “impressive” and other superlative adjectives have been removed in the revised manuscript.

Line 67... “are also involved in primary metabolism”. This statement is misleading in the way that primary metabolic enzymes can be a means to successfully generate genome mining predictions in a class-independent fashion. See EvoMining by Barona and co-workers. Not sure if EvoMining has been used for RiPP purposes but this needs to be clarified.

Response: We thank the reviewer for the suggestion. After consideration of the EvoMining workflow, we agree that the presence of genes in primary metabolism can aid the identification of new BGCs and have adjusted the statement accordingly. After a literature search, it appears that EvoMining has not been specifically applied to RiPPs, but it was discussed in a review article cited at the end of the paragraph (Hemmerling, F. & Piel, J. Strategies to access biosynthetic novelty in bacterial genomes for drug discovery. *Nat Rev Drug Discov*, 1-20 (2022)).

Line 150... “After dataset analysis....” Please clearly state in which way and what were the considerations for this selection, analysis.

Response: We thank the reviewer for the suggestion. The sentence now reads “After filtering the dataset for available strains and for BGCs encoding only primary modifying enzymes, we chose the representative from *Microbacterium paraoxydans* DSM 15019 for experimental characterization (Fig. 1).”

Line 153... “To validate the bioinformatics study...” If I am right, authors validate one single prediction and the entirety of the bioinformatics pipeline, analysis. Please change this.

Response: We thank the reviewer for the suggestion. The phrase has been altered to say “To examine the predicted BGC,” which limits the scope of the claims.

Line 255. “The gene deletion studies showed...” Please modify this to avoid confusion with real homologous gene deletion experiments. The *E. coli* experiments are combinatorial heterologous expression or BGC reconstitution experiments, which can lead to results that may differ with homologous systems.

Response: We thank the reviewer for the suggestion. Mentions of gene deletion have been altered to gene omission.

Line 302 – 318. This is the right place to discuss the hominycin observation, but I was wondering if a comparison between Gram positives and Gram negatives would also be interesting. This could be done both in terms of the distribution of the BGC (again Figure 1, complete) and/or the membrane-related MoA postulated.

Response: We thank the reviewer for the suggestion. The report of a likely hominycin BGC has been moved to the Discussion section in accordance with the recommendation. All but one of the daptide BGCs are encoded by Gram-positive organisms, including both daptides **1-3** and hominycin. Therefore, we are unsure what type of additional comparison would be useful.

Reviewer #3

Ren and colleagues report the discovery of the daptides, an unusual class of Ribosomally and Post-translationally modified Peptides (RiPPs) characterized by a novel C-terminal modified dimethylaminopropylamine (Dap). The authors pursued a unique in silico rationale for discovering new RiPP classes. Where previous studies mainly looked for biosynthetic enzyme homologs, the present study uses an alternative strategy searching for conserved recognition motifs for binding the RiPP's prepeptide leader sequence. Using their previously developed PRE-Finder algorithm, the authors assigned an impressive ~500 biosynthetic gene clusters (BGCs) to the daptide family, making this one of more abundant RiPP classes to date. The authors proceeded to characterize the BGC from *Microbacterium paraoxydans* as a type compound for the family. Using a refactoring approach, the BGC was heterologously expressed, leading to the identification of daptides 1-3, which contain the C-terminal Dap residue. Combining genetic engineered BGC variants and mass-spec analysis, the three-step conversion of the C-terminal threonine in the propeptide to Dap was elucidated. Because of the dual charged peptide ends, the authors hypothesized a bioactive role for the daptides in interacting with the membrane and the compounds were shown to demonstrate hemolytic activity.

I really enjoyed the paper, and have no further suggestions for its improvement. The experimental procedures seem solid and I think this is an important and exciting new discovery that would be of broad interest in the (RiPP) natural products field. Given the scarcity of enzymes to modify C-termini, this work holds promise for future applications peptide engineering.

Response: We highly appreciate the positive comments from the reviewer.

Reviewer #4 (Remarks to the Author):

The main limitation of this approach is obvious: It is restricted to those RiPP classes containing recognition elements (RRE), i.e., about the half of prokaryotic RiPP classes, as the authors claim. Moreover, although the main objective is to find new RiPP classes through genome mining, -thereby avoiding some intrinsic problems of the screening-based approaches-, the fact is that such C-terminal modification is not really new, as it had already been uncovered with homininin, which was discovered through bioactivity-guided screening.

Response: We appreciate the comments from the reviewer. While we agree that the C-terminal modification was not completely new, it is expected that the genome mining approach will find biosynthetic gene clusters producing molecules that have been discovered through bioactivity-guided isolation in the past, which has been used for decades and uncovered numerous natural products. On the other hand, using the RRE-based genome mining approach has revealed obvious advantages: the biosynthetic origin of Dmp (also renamed in the manuscript as suggested by the reviewer) was found for the first time, along with 483 BGCs encoding 1441 putative precursor peptides that manifest potential structural diversity of daptides. This highlights the complementarity of both approaches. Additionally, given the number of reported RRE domains (>50,000 proteins) and their distribution in known classes, use of the RRE as a bioinformatic hook should be a valuable window into new RiPP classes. Compared to polyketides or nonribosomal peptides, RiPP BGCs do not have a universal domain required for biosynthesis, so the RRE serves as a useful tool to identify further classes of RiPPs. We acknowledge that mining RRE domains will not be universal, but this approach represents a direct line toward undiscovered RiPP classes that currently elude discovery efforts.

To better understand the scope of this class-independent genome mining approach, it would be nice that authors show other **RRE families** found as output of this mining.

A few considerations before publication:

I think authors should include in the manuscript or in the supplementary material some data on the number and types of different "**RRE families**" found with this approach.

Response: We appreciate the comments from the reviewer. However, RRE-Finder has previously been used to analyze all proteins in the UniProtKB database. All retrieved RRE-containing proteins are available in the publication: Kloosterman, A. M., Shelton, K. E., van Wezel, G. P., Medema, M. H., & Mitchell, D. A. (2020). RRE-Finder: a genome-mining tool for class-independent RiPP discovery. *mSystems*, 5(5), e00267-20. Readers who are interested in using RRE for genome mining are directed to this publication for more details. Additionally, the algorithm for sorting the RRE domains has been provided in the supplementary information and online.

Given the Dap-C terminus modification is present in many distantly related BGCs, it appears to be a widespread PTM that may have emerged individually in each phylum (as the authors effectively say), rather than a class-defining feature in the strict sense.

Response: We appreciate the comments from the reviewer. However, among the 483 daptide BGCs identified through bioinformatic analysis, 482 of them are from either *Actinomycetota* or *Bacillota*. So we disagree with the reviewer that the Dmp-modified C-terminus would be considered a widespread PTM. In addition, using structures shared by products emerging from different phyla as class-defining features is also an acceptable paradigm in the nomenclature of RiPPs. For example, lanthipeptides are produced by numerous phyla, and class I-V lanthipeptide synthetases do not appear to have evolved from a common ancestor. Despite this, all these products fall into the same biosynthetic class. In the case of the narrower daptide class, this will likely be the case as well.

The authors found that all those BGCs contain an invariant Thr residue at the C-terminus. However, I could not find the corresponding “T” hallmark in the Skylign logo (supplementary figure 2).

Response: We thank the comment from the reviewer. The “T” hallmark was in light cyan which is not obvious. We have remade the Skylign logo and colored everything black to make the lettering easier to read. This modified version has been placed in the main text Fig. 1, while the unmodified version is kept in the supplementary information.

If only as a side comment, I think the abbreviation “Dap” used for naming the 2-(dimethylamino)propylamine moiety and the term “daptides” for the new class are confusing, since “Dap” is commonly used to refer to diaminopropionic acid or more rarely, diaminopimelic acid. The same moiety was abbreviated as “Dmp” in hominicin.

Response: We thank the reviewer for the comment. We have changed the name “Dap” into “Dmp” across the manuscript. The authors also changed the 2-(dimethylamino)propylamine to *N*₂,*N*₂-dimethyl-1,2-propanediamine.

The authors acquired 2D NMR spectra of 1 but only show summarized data for the Dap moiety in supplementary Table 2. In that table, carbon chemical shift of C-3 methyl is missing (“not available”), but in the HSQC spectrum of 1 (supplementary figure 14), a cross-peak at ca. 6-6.5 ppm appears to correlate with the methyl protons signal at 1.34 ppm. Is this signal due to any impurity? Did the authors avoid this assignment (as a genuine signal in 1) for any reason?

Response: We thank the comment from the reviewer. The assignment was originally avoided due to the relatively large difference in chemical shift compared to hominicin (4.0 ppm). However, as the reviewer suggested, we also assigned signals of the alanine residue that is next to the Dmp. Compared with the same penultimate alanine in hominicin, which is also attached to the Dmp, the chemical shift of methyl carbon in the daptide alanine also deviates from the hominicin one at a similar scale (2.7 ppm). Considering the NMR of daptide and hominicin are performed in different solvent, we hypothesized that such a deviation

of the chemical shift for the methyl carbon may be a solvent effect and decided to assign the C-3 with a chemical shift of 6.9 ppm with more confidence.

In addition, the CH-2 proton signal is indicated as “br s” but should be noted as a multiplet (m), even although it was an apparent broad singlet.

Response: We thank the reviewer for the comment. The annotation has been fixed.

NMR assignments of the Dap moiety (except that of C-3) are well supported and consistent with the structure proposed for daptide 1. The connectivity of Dap to the C-terminus of daptides was supported by HR-MS/MS in 2 and 3. However, the only spectroscopic evidence for connectivity between Dap and the rest of the peptide (1) is an HMBC correlation with an amide carbonyl at 175.8 ppm, which in turn was not assigned to any amino acid residue in the peptide.

I believe that full structural characterization (NMR assignments) of the compound should be done to better support the structure of these new RiPPs. I think the authors could have put more effort on this issue. If overlaps in NMR spectra of 1 would make it impossible to complete, perhaps the assignments of peptides 2 or 3 (smaller and with fewer redundant residues) might be easier.

Response: We thank the reviewer for their comment. NMR signals of the alanine residue next to the Dmp were fully assigned in the revised manuscript (Supplementary Fig. S14, Supplementary Table 3), and the connectivity to Dmp is suggested by HMBC correlations from the H-1' and H-2' to the carbonyl carbon at 175.8 ppm. In addition, assigned chemical shifts of the alanine residue in daptide 1 were close to the reported data of the Dmp-connected alanine in homininin.

We also attempted to assign other NMR signals to the proposed structure. Even though signals related to the aromatic structures in Trp-2 and Tyr-6 were also assigned (Supplementary Fig. S14), we were not able to assign the full structure of the compound due to missing and overlapping signals. In addition, considering the lower productivity and redundant residues in daptide 2 and 3, we expect the same issue in deducing their NMR spectra. Chemical synthesis of daptides is also challenging due to their high hydrophobicity. However, with the assigned NMR signals showing the Dmp structure and connectivity, together with the high-resolution MS/MS and Marfey's assay, the structure characterization of the daptides is solid.

6. As for the Marfey's analysis of 1, the authors did not find the FDAA adducts of Trp and Tyr, even although it was hydrolyzed under relatively mild conditions (DCI, 95 C, 2 h; 3% phenol). Trp degradation is a common issue but that of Tyr is rarer under these conditions. Did the authors interrogate the LC-MS chromatograms for both mono- and di-FDAA derivatives of Tyr?

Response: We thank the reviewer for their comment. Unfortunately, single ion monitoring was used on the original sample, so possible di-FDAA derivatives were not observed. This is because previous work within the lab under these conditions did not require monitoring of the di-FDAA adducts, and the minimal number of ions were selected for monitoring to maximize sensitivity on the available LC-MS instrument. After receiving your comment, we opted to add 686 m/z for monitoring in our methods, and we were able to identify the FDAA(bis)-Tyr species in our standards. Our hydrolysate samples then showed FDAA(bis)-L-Tyr adducts in both the mpa and sca cases. Text, methods, and figures have been edited to show the assignment of Tyr.

7. In Figure 3b, the MALDI-TOF MS analysis of the mpaABCDM full construct shows the presence of intermediates 5/6 and the final product 7, but also a compound at lower m/z (unlabeled in the figure). Any idea of the identity of this species?

Response: We thank the reviewer for the comment. The signal at the lower m/z is 54 Da smaller than the unmodified precursor peptide. We attempted to locate the putative modification by peptidase digestion but weren't able to get conclusive results, most likely owing to poor solubility of full length MpaA1 in the highly aqueous digestion conditions. It is possible that the signal might be a MALDI artifact of the Dmp-modified peptide, but we have no concrete evidence as to the signal's identity.

8. "Expression of mpaABC (omission of mpaM and mpaD) resulted in the accumulation of the ketone 5, consistent with transaminase activity for MpaD".

The omission of only mpaD should have led the same result. Why did the authors simultaneously omit both mpaM and mpaD?

Response: We appreciate the comment from the reviewer. We have constructed the plasmid with only mpaD omitted which led to the same result as that with omission of both mpaM and mpaD. The MALDI-TOF-MS result and the plasmid map were updated in the revised manuscript (Fig. 3, Supplementary Fig. 41), and the single omission of mpaD is now explicitly mentioned in the main text alongside the double omission of mpaD and mpaM.

REVIEWERS' COMMENTS

Reviewer #1 (Remarks to the Author):

All my comments have been addressed appropriately and I think the manuscript improved even more. I think this is a great story.

Reviewer #2 (Remarks to the Author):

Authors did a good job and even if I do not agree with some of their “believes” I am Ok with the paper in its current form. I am happy with Figure 1 and 4 and the fact that they did some extra experiments, as suggested by Reviewers 2 and 4.

Reviewer #4 (Remarks to the Author):

I thank the authors for the revised manuscript and supplementary material. All my comments have been addressed and reflected in them